# Synthetic seismicity distribution in Guerrero-Oaxaca subduction zone, México and its implications on the role of asperities in Gutenberg-Richter Law

Marisol Monterrubio-Velasco[1], F. Ramón Zúñiga[2], Quetzalcoatl Rodríguez-Pérez[2,3], Otilio Rojas[1,4], Armando Aguilar-Meléndez[1,5], and Josep de la Puente[1]

[1]Barcelona Supercomputing Center - Centro Nacional de Supercomputación, Jordi Girona 29, 08034, Barcelona, Spain
[2]Universidad Nacional Autónoma de México, Centro de Geociencias, Juriquilla, Querétaro, México
[3]Consejo Nacional de Ciencia y Tecnología, Mexico City, 03940, México
[4]Escuela de Computación, Facultad de Ciencias, Universidad Central de Venezuela, Caracas, 1040, Venezuela
[5]Facultad de Ingeniería Civil,Universidad Veracruzana, Poza Rica, Veracruz, 93390, México

**Correspondence:** Marisol Monterrubio-Velasco (marisol.monterrubio@bsc.es)

**Abstract.** Seismicity and magnitude distributions are fundamental for seismic hazard analysis. The Mexican subduction margin along the Pacific Coast is one of the most active seismic zones in the world, which makes it an optimal region for observation and experimentation analyses. Some remarkable seismicity features have been observed on a subvolume of this subduction region, suggesting that the observed simplicity of earthquake sources arises from the rupturing of single asperities. This sub-region has been named SUB3 in a recent seismotectonic regionalization of Mexico. In this work, we numerically test this hypothesis using the TREMOL (*sThochastic Rupture Earthquake MOdeL*) v0.1.0 code. As test cases, we choose four of the most significant recent events ($6.5 <$ Mw $< 7.8$) that occurred in the Guerrero-Oaxaca region (SUB3) during the period 1988-2018, and whose associated seismic histories are well recorded in the regional catalogs. Synthetic seismicity results show a reasonable fit to the real data, which improves when the available data from the real events increases. These results give support to the hypothesis that single asperity ruptures are a distinctive feature that controls seismicity in SUB3. Moreover, a fault aspect ratio sensitivity analysis is carried out to study how the synthetic seismicity varies. Our results indicate that asperity shape is an important modeling parameter controlling the frequency-magnitude distribution of synthetic data. Therefore, TREMOL provides appropriate means to model complex seismicity curves, such as those observed in the SUB3 region, and highlights its usefulness as a tool to shed additional light on the earthquake process.

## 1 Introduction

The variation in seismicity distributions for different regions is a key input for Probabilistic Seismic Hazard Analysis (PSHA), as well as for other hazard determination approaches. The frequency-magnitude distribution from individual faults determines

the specific earthquake rate of a given size at each source point, which has an important influence on the PSHA outcome
(Cornell, 1968; Parsons et al., 2018; Main, 1995). To estimate the earthquake frequency in a given region and time span, the
linear relation of the frequency-magnitude distribution known as Gutenberg-Richter (GR) law, is one of the most employed
empirical relations in seismology,

$$\log_{10} N(\geq M) = a - bM \,, \tag{1}$$

where $N$ is the cumulative number of earthquakes greater than a specific magnitude $M$. The parameters $a$ and $b$ depend on
regional tectonic characteristics, such as the seismicity level and the stress regime (Ozturk, 2012; Evernden, 1970). Despite the
fact that the GR distribution is widely used, other distributions have also been employed to describe frequency-magnitude ob-
servations. For example, paleoseismological data suggest that a specific fault segment may generate characteristic earthquakes,
causing an increase in preferred magnitudes, as observed in California (Parsons and Geist, 2009; Schwartz and Coppersmith,
1984) or Japan (Parsons et al., 2018). A characteristic earthquake model implies a non-linear earthquake frequency-magnitude
distribution, highly dominated by the occurrence of a preferred size event that induces low $b$ values, or plateaus (Schwartz and
Coppersmith, 1984; Wesnousky et al., 1983). In such cases, a GR relation is not a good representation, and therefore it is not
appropriate to describe the earthquake frequency relation for those particular regions (Aki, 1984; Parsons et al., 2018). Alterna-
tively, depending on the regional tectonics, the size distribution of earthquakes could generate a "mixed" frequency-magnitude
distribution (Lay et al., 1982; Dahmen et al., 2001), where the frequency-magnitude fits a GR distribution at intermediate
magnitudes, but large events (associated with the characteristic earthquake) depart from a linear GR relation (Lomnitz-Adler,
1985; Dalguer et al., 2004).

Some authors have provided a possible explanation of the physics underlying the earthquake process observed in the tran-
sition from a GR-type to characteristic-type behavior. For example, Wesnousky et al. (1983) pointed out that while regional
seismicity satisfies the GR relation, the seismicity on individual faults does not. They also suggest that GR model may not be
applicable to an individual fault or fault segment, and they consider the model proposed in Allen (1968) as an explanation to
those cases. Allen (1968) comments that the fault segments that generate earthquakes of a characteristic size, where the absence
of moderate and small earthquakes occurs, is a function of fault length and tectonic setting. Moreover, Wesnousky et al. (1983)
in their conclusions say that the size of the characteristic event is a function of the fault length, and the frequency-magnitude
distribution particular to a single fault does not satisfy the GR relation. On the other hand, Stirling et al. (1996) also stated that
different studies have reported evidence to suggest that seismicity along faults does not satisfy GR-type distribution, across
the entire magnitude range. Instead, seismicity along faults shows a greater frequency of occurrence of large earthquakes than
would be expected from an extrapolation of curves fit the log-linear distribution of lesser-sized earthquakes. Moreover, they
also comment that determining whether it is the GR relationship or the characteristic earthquake model that describes the
seismicity along particular faults is problematic because historical records of seismicity are generally much shorter than the re-
currence time of the largest earthquake on a fault. On the other hand, Leonard (2010) proposes a possible physical explanation
to answer why the characteristic model occurs. He comments that once a fault's width approaches the width of the seismogenic
zone, strike-slip earthquakes become fixed width and the fault expands only in length. In this sense an interesting discussion is

also found in Sibson (1989). He pointed out some important questions about the earthquake faulting and the structural geology. In fact, he commented that earthquakes and related processes are an integral part of structural geology. In particular, he proposes that the fault segmentation leading to characteristic earthquake behavior demands the existence of persistent structural controls at segment boundaries, governing the nucleation and arrest of ruptures. Moreover, the fault aspect-ratio, $\chi$, defined as the effective fault length over the effective fault width, has been found to play a crucial role in empirical and numerical studies. For example, Heimpel (2003) conducted a study via numerical simulations about the variations of frequency-magnitude distributions due to changes of the aspect ratio. They found that for large $\chi$, i.e., thinner rectangle faults, quasi-periodical ruptures break the entire fault, and smaller events do not occur. They attribute these observations to the characteristic length of large events, in their models, have an aspect ratio approaching that of the entire fault. In the case of a strike-slip fault type, Heimpel (2003) found larger values of $\chi$. The seismogenic widths of strike-slip faults are usually less than 20-30 km according to the finite fault rupture models of earthquakes (Weng and Yang, 2017). Tejedor et al. (2009) argue that the aspect-ratio of real faults is important because it appears that has a direct relation with the overall size of the fault plane. Specifically, small faults usually being a square aspect ratio, $\chi \approx 1$, whilst big faults being elongated with aspect ratio, $\chi > 1$, due to the depth limit that the brittle-ductile transition imposes on the Earth crust (around 15 km for vertical strike-slip faults, and twice that for subduction-type faults). Considering this depth limit and the range of surface fault trace lengths (from few kilometers in the case of small earthquakes to hundredths of km for great earthquakes), based on observations Weng and Yang (2017) analyzed and reported different aspect-ratio values for strike-slip and dip-slip earthquakes. In general, the aspect-ratio values are in the interval $1 < \chi < 8$, excepting for some strike-slip events that could reach larger values $\approx 40$. In our work, we refer to the aspect ratio of the asperity within a fault, not the proper fault geometry. It is still true that the asperity might grow in length as opposed to width for near vertical faults once they reach the brittle limit of the crust, however, we assume that this would be an extreme case. Stock and Smith (2000) shows that the aspect-ratio of dip-slip earthquakes is similar for all earthquake sizes. Hence, the limitation in rupture width seems to control the maximum possible rupture length for these events. They also found, after analyzing five cases of real earthquakes, that only one normal event (31/6/1970, Columbia, $M_w = 7.7$) and four reverse events (09/03/1957, Aleutians, $M_w = 8.25$; 22/05/1960, Chile, $M_w = 8.5$; 04/02/1965, Rat Island, $M_w = 8.25$; 29/09/1973, Vladivostok, $M_w = 7.75$) have a rupture length to width ratio larger than 4.

Studies on the frequency-magnitude distributions of earthquakes in the Pacific subduction regime of Mexico are not extense. Singh et al. (1983) report that the GR relation was not appropriate to model the occurrence of large earthquakes in the Mexican subduction zone. They found that the GR relation in the range $4.5 \leq M_s \leq 6.0$, when extrapolated, grossly underestimated the observed frequency of large earthquakes ($M_s \geq 6.5$) for the Oaxaca and Jalisco regions. In that sense, significant efforts are oriented towards understanding, in great detail, the properties of the seismic regions as they influence the distributions of seismicity. In particular, the subduction regime along the Pacific Coast of Mexico is a region where earthquakes of relevance in terms of damage (e.g. Mw > 6.0) take place quite frequently (see Fig. 1 (a)). The most recent devastating cases, the 1985 $M_w$ = 8.1 Michoacan earthquake, that killed more than 20,000 people, and the 2017 $M_w$ = 8.2 Puebla-Morelos earthquake, which had more than 100 casualties and at least 300 injured, are two dramatic examples. As a consequence, this region is the main contributor to the seismic hazard of Mexico, although other regions also play an important role (Yazdi et al., 2019).

In this context, Zúñiga et al. (2017) recently proposed a seismotectonic regionalization of Mexico with the purpose of hazard and risk assessment. Among other regions, the authors defined as SUB3 an area located in Guerrero-Oaxaca states, as one of the subregions in the subduction regime. SUB3 zone presents the following two characteristics. First, seismicity corresponds to the shallow (h < 40 km) strong coupling of subduction, covering the transitional zone of the Cocos - North American plates convergence. Second, it evolves along a plate boundary with simple and homogeneous fault surfaces, where slip takes place on single asperities (see also Singh and Mortera, 1991). These two features are apparent in the frequency-magnitude cumulative curve as characteristic events, which do not obey the linearity of the GR law. These authors described the singular frequency-magnitude relation depicted in Fig. 2 (a), based on data that spans the period 1988-2014. This distribution shows a highly distinctive feature, such as an abrupt change of the frequency-magnitude tendency at the magnitude range $6.4 \leq M \leq 7.3$. In Fig 2 (b), we show the magnitude histogram of the SUB3 region for the period of 1988-2020, as a way to stress the lack of events in the range $6.0 < M < 6.4$. This behavior has been interpreted as the result of seismic events rupturing similar asperities. These repeating earthquakes of similar magnitudes have been identified as "characteristic" events of that volume (Singh et al., 1983). During the past 100 years large events were registered in this area, such as the events in April 15th, 1907 ($M_s$ 7.7); March 26th, 1908 ($M_s$ 7.6); June 17th, 1928 ($M_s$ 7.8); October 9th, 1928 ($M_s$ 7.6); December 23rd, 1937 ($M_s$ 7.5); July 28th, 1957 ($M_s$ 7.7); August 23rd, 1965 ($M_s$ 7.8); October 29th, 1978 ($M_w$ 7.8); and March 20th, 2012 ($M_s$ 7.5). These earthquakes were strongly perceived in cities like Acapulco, Oaxaca, and Mexico, causing significant damages in some cases. As was proposed by Zúñiga et al. (2017) and Singh and Mortera (1991) one explanation for this behavior of the SUB3 region is related with the rupture of single asperities. As referred in the classical literature, asperities correspond to strong patches that are resistive to breaking and release a larger amount of seismic moment during subsequent ruptures (Aki, 1984; Lei, 2003; Rodríguez-Pérez and Zúñiga, 2017).

Following Somerville et al. (1999b) asperities are defined as regions of irregular shape on the rupture plane at which slip is 1.5 or more times larger than the average slip. Accordingly, Rodríguez-Pérez et al. (2018) use finite-fault solutions reported for the Mexican subduction zone to estimate effective dimensions, average displacement, the combined asperity area to effective rupture area ratio, among other parameters. Rodríguez-Pérez et al. (2018) also points out that characterizing asperities at interfaces is crucial for seismic hazard analysis, because during ruptures these zones would suffer the highest stress drops and slip values. Therefore, ruptures in these areas may generate the strongest ground motion. Ruff (1992) determined the distribution of major asperities along plate boundary segments for several subduction zones, as the Kurile Islands, Colombia, and Peru subduction zones. Also, Yamanaka and Kikuchi (2004) carried out an analysis and characterization of the asperities that produce strong earthquakes in the subduction zone in northeastern Japan.

Considering the aforementioned observations on the seismicity of the Mexican SUB3 region, the motivation of this work is to present an alternative way to analyze the influence of the asperities on the frequency-magnitude distribution of that region. This study uses the *sTochastic Rupture Earthquake MOdeL* (TREMOL) scheme (Monterrubio-Velasco et al., 2019a), that is a specialized code for the simulation of earthquake ruptures. Earlier results (Monterrubio-Velasco et al., 2019a) showed that this numerical model appropriately simulates the maximum magnitudes observed at the Mexican subduction zone. Altogether, TREMOL has also shown flexibility to simulate different scenarios with few parameters, as in the case of aftershocks follow-

ing predefined faults (Monterrubio-Velasco et al., 2019b). An important TREMOL addition to the modeling parameters is the inclusion of asperities along the fault plane. Moreover, an additional objective is to complement our magnitude distribution analyses, by also exploring the influence of the fault aspect-ratio on the synthetic seismicity. This analysis arises from ob-
125 servations of the relevant contribution of this fault parameter on the magnitude distribution characteristics studied in different tectonic regions (Weng and Yang, 2017; Yoder et al., 2012; Stock and Smith, 2000; Main, 2000, 1995; Main and Burton, 1989). To our knowledge the present work is the first stochastic model based on Fiber Bundle approach that simulates the frequency-magnitude distribution and its likely dependence on the source aspect-ratio.

As already mentioned above, we focus in this work on the Guerrero-Oaxaca SUB3 given that this region provides an ideal setting for testing the single asperity paradigm with the aid of TREMOL. Moreover, the quality of the database allows us to validate our code, giving support to the extension of our numerical experiments to other regions where few registered earthquakes due to scarce seismic networks. In this sense, our study pretends to be useful to generate synthetic seismicities to allow completing earthquake databases, in order to carry out more accurate PSHA studies. We also consider that our study
could be appropriate to study different configuration of seismic scenarios, such as the occurrence of large past events that lack records, or future events with a significant hazard as the case of the Guerrero gap.

## 2 TREMOL

TREMOL is a numerical method for the simulation of the earthquake rupture process able to contemplate different seismic scenarios. In this work TREMOL starts with the occurrence of previous low-magnitude events and culminates with the main-
140 shock. The current TREMOL implementation does not allow simulating a full earthquake cycle, because most of the tectonic load is spent during the whole process of the mainshock rupture and foreshocks, and no extra load is added during the simulation (Monterrubio-Velasco et al., 2019a). TREMOL is based on the Fiber Bundle Model (FBM) that describes the rupture process in heterogeneous materials (Hansen et al., 2015). The FBM analyzes the earthquake dynamics from the point of view of deformable materials that break under critical stresses. An implication of the FBM is the self-organized criticality behavior
of the rupture process from micro to macro scale (Pradhan et al., 2010; Aki, 1984; Lei, 2003; Rodríguez-Pérez and Zúñiga, 2017). Among its main assumptions, TREMOL considers the existence of a main asperity in the seismic region, associated to the modeled maximum-magnitude earthquake. Furthermore, this asperity is assumed to have a rectangular shape with a predefined aspect ratio. Given the relevance of this single-asperity hypothesis, we define as a "Single-Asperity region", or SA region, as a rupture zone that contains a large single asperity, that experiences the largest slip during the modeled mainshock.
This asperity belongs to an effective fault area, not precluding the occurrence of other previous smaller events. As mentioned above, the observed seismicity in the Mexican SUB3 subduction region can be assumed to be controlled by single asperity contacts, and therefore, we consider TREMOL as a suitable modeling tool to study such processes.

As described by Monterrubio-Velasco et al. (2019a), TREMOL makes use of few input parameters for a complete definition of a SA region. In particular, the following four parameters are required for a general finite-fault discrete model:

1. The effective length $L_{\text{eff}}$ $[km]$.

2. The effective width $W_{\text{eff}}$ $[km]$.

3. The asperity size $A_{\text{a}}$ $[km^2]$, defined for each SA region.

4. The discrete number of cells $N_{\text{cell}}$ that defines the size of the computational domain.

In addition, the following TREMOL parameters allow setting the load and fault strength distributions, in addition to asperity features:

5. The load conservation parameter $\pi$. After a cell fails, the TREMOL algorithm transfers its load to the neighbor cells, keeping a 1-$\pi$ portion of its initial load. Two parameter values of $\pi$ are defined at the asperity level and at the remaining background area, named $\pi_{\text{asp}}$ and $\pi_{\text{bkg}}$, respectively.

6. The asperity strength value $\gamma_{\text{asp}}$. Since, the asperity shows a physical resistance to break, this parameter allows controlling the "hardness" of the asperity, and therefore its capability to break.

7. The ratio of the asperity area $S_{\text{a}-\text{Asp}}$, that is computed as,

$$S_{\text{a}-\text{Asp}} = S_{\text{a}} + 0.5(S_{\text{a}} \cdot \alpha) \,, \tag{2}$$

where $\alpha$ is a random number, and $S_{\text{a}} = A_{\text{a}}/A_{\text{eff}}$. This parameter allows setting a random size of the modeled asperity, that mimics the uncertainty and variability of real values (Somerville et al., 1999b; Murotani et al., 2008; Blaser et al., 2010; Strasser et al., 2010).

It is worth mentioning that the strength $\gamma$ and the load-transfer $\pi$, are our modeling devices of the physical properties of rock hardness and radiated energy, respectively.

The TREMOL workflow is summarized in three main stages: (i) A pre-processing stage where the input parameters are set, (ii) A processing stage that performs the FBM simulation of the whole rupture process and, (iii) A final post-processing that converts output results into a synthetic seismic catalog. During processing, TREMOL generates numerous smaller earthquakes until the rupture of the whole asperity area $S_{\text{a}-\text{Asp}}$ is achieved. In the post-processing stage, TREMOL also calculates the actual area ruptured during each earthquake, and reports such area in physical units $[km^2]$ to allow comparisons with the whole effective area. Notice that it is possible to associate various magnitude values to the same final earthquake area, by using alternative magnitude-area relations. These magnitude values may present a strong variability with a significant impact on the synthetic seismicity curve, and then the selection of a magnitude-rupture area relation is a crucial hypothesis of this kind of studies. In this work, we use four magnitude-area relations following those proposed by Rodríguez-Pérez and Ottemöller (2013), i.e,

$$M_{wS} = (\log_{10} A_{\text{a}} + 4.393)/0.991 \,, \tag{3}$$

$$M_{wML} = (\log_{10} A_{\mathrm{a}} + 5.518)/1.137, \tag{4}$$

$$M_{wMVL} = (\log_{10} A_{\mathrm{a}} + 6.013)/1.146, \tag{5}$$

and the one proposed by Ramírez-Gaytán et al. (2014), specifically developed for subduction events in México

$$M_{wR} = (2/3) * (\log_{10} A_{\mathrm{a}}/(7.78 * 1.0e - 9)^{(1/0.550)}) - 6.07. \tag{6}$$

In these equations, $A_{\mathrm{a}}$ is the area [km$^2$] of each earthquake generated in the seismic region or domain $\Omega$ according to the nomenclature of TREMOL (Monterrubio-Velasco et al., 2019a). Relation in Eq. 3 was obtained from asperities defined by the average displacement criterion (Somerville et al., 1999a). Relations in Eqs. 4 and 5 were found by using the maximum displacement criterion for a large and a very large, asperity, respectively (Mai et al., 2005).

Moreover, in order to compare the results obtained by the magnitude-area relations we also estimate the magnitude from the moment-magnitude relation given in Aki and Richards (2002) as

$$M_w = 2/3 log_{10}(Mo) - 6.07 \tag{7}$$

where $Mo$ is computed following Kanamori and Anderson (1975) relation

$$Mo = 16/7\Delta\hat{\sigma} r^3 \tag{8}$$

where $\Delta\hat{\sigma}$ is the stress drop and $r$ is the radius of the fault. In our model $r$ is computed from the rupture area of each synthetic earthquake estimated as a circular area. $\Delta\hat{\sigma}$ is obtained from the SUB3 region database (Rodríguez-Pérez et al., 2018). Fig. 3 shows the spatial distribution of the earthquakes used to determine the stress drop. A large stress drop dispersion is observed in this region as is shown in Fig. 3 (b). Therefore, we use the mean and median values in Eq. 8, being $\Delta\hat{\sigma} = 9.46$ MPa and $\Delta\hat{\sigma} = 1.42$ MPa, respectively.

TREMOL is capable of estimating the rupture areas assigning physical units to the numerical domain. In this paper, we do not consider slip to compute the magnitude distributions. On the other hand, TREMOL is not able to model the stress drop since the tectonic load is simulated using dimensionless units. We estimate a mean load drop, not related to any physical unit.

To determine the seismicity curve of a given SA region, TREMOL computes, as a part of the postprocessing, the frequency-magnitude distribution associated with this region. In the case of the SUB3 region, Zúñiga et al. (2017) discuss the singular behavior followed by the observed frequency-magnitude distribution.

## 3  Essential data

As basic testing data, we use four subduction earthquakes which occurred in the SUB3 region, from the database published by Rodríguez-Pérez et al. (2018). Hereafter, this database is referred to as "DB-FiniteFault-2018". We use these events because

their epicentral coordinates fall into the SUB3 region, their magnitudes are greater than 6.0, and they occurred after 1988
which is a date that indicates the start of the most homogeneous recording conditions of the network (Zúñiga et al., 2012).
The epicentral location and the necessary seismic information of these four mainshocks are shown in Fig. 1 (c) and Table 1,
respectively. The spatial representation of the effective area associated with these earthquakes is shown in Fig. 1 (c).

It is important to emphasize that, according to results in Rodríguez-Pérez and Ottemöller (2013), the size of an earthquake
not only depends on the effective area. It also depends on the size of the asperity, among other possible influential variables.
For example, in the case of the four earthquakes reported in Table 1 the maximum effective area is equal to 3488.52 km$^2$ and is
associated to an earthquake of magnitude 7.1, according to Table 2. However, the mainshock with the largest magnitude (7.8)
is associated to a smaller effective area of 3086.22 km$^2$.

## 4   Methodology

The SUB3 region is approximately delineated by the polygon shown in Fig. 1 (b). To validate our synthetic results, we extract
earthquake (magnitude and epicenter) data from the catalog of the Mexican SSN (2019) from 1988. Hereafter we refer to this
catalog as "SSN-1988-2018". In order to numerically test the main hypothesis of this work, namely that the seismicity of SUB3
as mainly originated from ruptures at single asperities, we apply the following global framework.

### 4.1   Global TREMOL framework: single asperity and aggregated curves

1. Using the database "DB-FiniteFault-2018", we identified all earthquakes with a magnitude greater or equal than 6.5
and occurred within the SUB3 region after 1988 (Table 1). As a result, only four mainshocks satisfy these criteria
whose hypocenters are illustrated in Fig. 1 (b). However is worth noting that in this region and during this period more
earthquakes with a magnitude greater or equal than 6.5 have been generated. However, we are not considering them
because they are not integrated in "DB-FiniteFault-2018" due to lack of source parameters estimations. Notice that each
of these earthquakes has associated a maximum asperity area, which at the same time, defines a SA region of size $A_{\text{eff}}$,
depicted in Fig. 1 (c) (as described in subsection 2).

2. We apply TREMOL v0.1.0 to simulate the seismic activity at each SA region. Even though these SA regions are depicted
as simple rectangles in Fig. 1 (c), the fault dip and epicentral depth are implicitly considered in TREMOL simulations
since the synthetic activity can be seen as a 3-dimensional projection into a bidimensional plane. The dates of the
mainshocks and duration of associated seismicity are well separated in time, so that we can consider each SA region as
independent for an individual TREMOL simulation.

3. We finally add the four individual synthetic curves to obtain an aggregated seismicity curve for the study area. This area
corresponds to 15%-20% of the SUB3 region, approximately. Is worth mentioning that TREMOL 0.1.0 does not model
the simultaneous interaction among the four sources, i.e., the Coulomb stress changes from one source to the next are not

considered. However, the objective of this exercise is to aggregate the curve as an example of the aggregated seismicity
without considering the interaction between sources. Future TREMOL generalizations would include such interactions.

In the upcoming sections, we describe further details of the simulation procedure based on TREMOL, and we base our
discussion on comparisons of synthetic results with observed seismicity.

## 4.2 Input parameters and realizations

A TREMOL simulation of each one of the four mainshock earthquakes given in Table 1, requires the values of $L_{\text{eff}}$ and $W_{\text{eff}}$,
also given there, as well as the asperity size $A_{\text{a}}$ of each SA region, that is easily determined from $S_{\text{a}}$ and $A_{\text{eff}}$. A few additional
input parameters are specified below. An important consideration relates the uncertainty quantification on the real size of the
large and single asperity at each study region. We perform 20 realizations at each SA region by changing the random parameter
of the modeled asperity size given by Eq. 2, and then we average these results. The number of 20 realizations was chosen
because we estimated by statistical testing that the standard error is invariant for more than 5 realizations, so we consider that
20 is enough to provide a robust statistical outcome. The following are the steps for our numerical test:

1. Defining the input model parameters required by TREMOL. In addition to values given in Table 1, TREMOL also
   employs as parameters $\pi_{\text{asp}} = 0.90$, $\gamma_{\text{asp}} = 4$, and the total number of cells $N_{\text{cell}} = 40000$ to define the model domain
   and characteristics. The number of cells represents a $N_{\text{x}} \times N_{\text{y}}$ discretization of the fault plane that follows its real aspect
   ratio given by $\chi = L_{\text{eff}}/W_{\text{eff}}$ due to the relations,

$$N_{\text{y}} = \sqrt{\frac{N_{\text{cell}}}{(W_{\text{eff}}/L_{\text{eff}})}}, \quad \text{and} \quad N_{\text{x}} = (W_{\text{eff}}/L_{\text{eff}}) \times N_{\text{y}}, \tag{9}$$

   where $N_{\text{x}}$ and $N_{\text{y}}$ are the number of cells along the horizontal and vertical direction, respectively.

2. As statistical support to our resulting curves, we execute TREMOL v0.1 twenty times per each SA region listed in Table
   1. At each execution, the asperity size is modified according to Eq. 2. At the end of each realization, the rupture area of
   each synthetic event is customarily calculated by TREMOL, and then its equivalent magnitude is computed using Eqs.
   3, 4, 5, and 6.

3. For each realization, we also compute the frequency-magnitude distribution of synthetic earthquakes. To do so, we split
   the magnitude range $M_w \in [2.5, 9]$ into 65 subintervals and count the number of these synthetic events at each magnitude
   bin. Once the twenty executions for a single SA region have been completed, we also compute the mean and standard
   deviations of the number of earthquakes at each magnitude bin.

4. Finally, after the four SA simulation sets have been computed, we add their contribution, in the frequency-magnitude
   range, to the aggregated seismicity curve, considering their mean and standard deviation. This global curve represents
   the synthetic seismicity of a seismic area about 15%-20% of the whole SUB3 region.

### 4.3 Observed seismicity distribution

As basis for comparison for TREMOL output, we compiled the distribution of seismicity from a seismic catalogue SSN-1988-2018 of 34716 events that occurred at the SUB3 region from 1988 to 2018 with a minimum magnitude of 1.5 $M_w$. We extracted from this catalog the events that satisfy the following criteria:

1. The epicentral latitude and longitude coordinates must be within the study regions, according to Fig. 1 (a).

2. They should fall within the reference depth which corresponds to the mainshock hypocenter depth. We included all events in a range of 8 km above and below the mainshock depth to account for the uncertainty on this value, which is a well-known limitation on the hypocentral location. Moreover, in the case of the 25/02/1996 earthquake, we considered all events regardless of their depth, because of the lack of data in the reference catalogue (SSN-1988-2018).

3. The occurrence time should fall into the temporal window given by the catalog start date (1/1/1988) and until half a year after the corresponding mainshock date.

The above selection criteria agree with the phenomenology simulated by TREMOL, which aims to model the previous seismic activity up to the mainshock, and in some cases, a few events just after its occurrence since the simulation ends when the area of $S_{a-Asp}$ is completely activated (ruptured). In this version, TREMOL simulations are limited to bidimensional domains (modeling a dipping fault plane), hence we necessarily have to use a hypocentral depth range for event acceptance. Thus, our consideration to include events $\pm 8$ km from the hypocenter depth is reasonable.

Lastly, it is worth mentioning that, when we construct the aggregated curve of the observed seismicity on the four SA regions of Fig. 1, we take into account each event only once, in the case of the overlapping areas, such as SA regions 1 and 4, or for the case of SA regions 3 and 2.

## 5 Results

### 5.1 Synthetic seismicity distributions

We obtained four synthetic curves computed at each SA region according to the four area-magnitude relations (Eqs. 3, 4, 5, and 6). We also show the synthetic curves obtained by the moment-magnitude relation (Eq. 7) using both the mean and the median of the stress drop of observed events as previously described. In Figs. 4, 5, 6, and 7 the blue line represents the real curve obtained from the earthquake catalogue referred in Section 4.3 (SSN-1988-2018), and black curves correspond to synthetic results, where the mean curve of the twenty TREMOL realizations is shown by the solid line, while the dotted lines indicate the standard deviation.

The SA region 1 has approximately an area equal to 3207 $km^2$ (Table 1), and Fig. 4 shows the observed and TREMOL synthetic seismicity curves. Each subplot in this figure shows a synthetic frequency-magnitude curve obtained for a particular area-magnitude relation and moment-magnitude (Eqs. 3, 4, 5, 6, 7). In the case of Fig. 4 (a), (b), (c), and (d) thus, differences

between these four seismicity curves are only attributable to the alternative area-magnitude relations used as a basis in their computation. And in the case of Fig. 4 (e), (f) the observed differences are due to the used stress drop values. Related to the magnitude-area relations, in this particular region, Eq.6 leads to the frequency-magnitude curve that best matches the observed one. The best fit in this curve is for magnitudes greater than 4.0. The second-best approximation corresponds to the case of Eq. 3, whose fit improves for magnitudes greater than 5.5. The third best fit is achieved by Eq. 4 for magnitudes greater than 6, and finally, the worst approximation is given by Eq. 5 with reasonable results only for magnitudes near 7. Once we compare the results with the moment-magnitude relation we observe that using the median stress-drop the synthetic curve shows a good fitting from magnitude larger than 4.0. However, the estimated maximum magnitude with this relation is underestimated (Fig. 4 (f)). On the other hand, using the mean stress drop value, the curve moves shifts to larger magnitudes. This shifting better fits the maximum magnitude but overestimates lower magnitudes (Fig. 4 (e)).

A similar analysis can be done for the other SA regions. According to the results for region 2 in Fig. 5, we observe that the best fit is obtained by the application of Eq. 6. In this case, the synthetic seismicity curve closely approaches the observed data for magnitudes greater than 4.0. Comparing the synthetic curve obtained by using the median stress drop in Eq. 7 we observe a good fitting for magnitude $M < 6$. However, as in the previous region, the maximum magnitude is underestimated

Results for region 3 depicted in Fig. 6, indicate that the best synthetic fit is achieved by the curve computed from Eq. 3, particularly, for magnitudes greater than 4.5. We can also observe that the poorest fit is obtained by using Eq. 6, which underestimates the real seismicity. The mean stress drop in Eq. 7 provides the best match to the observed distribution, however the computed maximum magnitude is low in comparison to that obtained from the magnitude-area relations.

Finally, TREMOL′s results for region 4 in Fig. 7 reveal the excellent fit attained by the application of Eq. 6. This latter result is the best fit overall, but we highlight the fact that this region includes the largest number of observed events. The median stress drop also generates a synthetic curve with results which match the observed data for magnitudes $M < 6$. Nevertheless, as in previous results, maximum magnitude is underestimated.

## 5.2 Synthetic aggregated curves

Aiming at approximating the seismicity of nearly 15%-20% of the SUB3 region, as mentioned in Section 4, we added the four synthetic SA curves previously computed into an aggregated single curve. We consider this part of the analysis a useful validation of our methodology based on TREMOL, providing important insight into the hypothesis of single asperity ruptures and its relation with real cases. Fig. 8 shows the aggregated seismicity curves, each one based on results corresponding to a particular magnitude-area relationship (Eqs. 3 - 6). In this figure, we observe that the synthetic curve based on Eq. 6, more closely matches the real seismicity curve for magnitudes larger than 4, while the other scaling relations only approach the observed seismicity curves for M >6. In Fig. 9 we plot the magnitude histogram computed from the aggregated seismicity using the Ramírez-Gaytán et al. (2014) relation. Gray and green bars indicate the standard deviation and red bars the mean values over 20 realizations per SA region. Comparing this figure with the real histogram in Fig. 2 (b), we observe similar characteristics for the magnitude range of $6 < M < 7$ with a clear decreasing of events, and an increasing for $M > 7$ earthquakes. This increment is related with the rupture of the single asperities.

## 6 Effects of the aspect ratio $\chi$ on the frequency-magnitude distributions.

In what follows, we discuss the sensitivity of the model to the aspect ratio $\chi = L_{\text{eff}}/W_{\text{eff}}$ as reflected in the shape of the F-M distributions.

Previous observational and numerical studies have implied a direct relation of the fault aspect ratio over the frequency-magnitude distribution (Weng and Yang, 2017; Yoder et al., 2012; Stock and Smith, 2000; Main, 2000, 1995; Main and Burton, 1989; Console et al., 2015; Tejedor et al., 2009; Heimpel, 2003). These works motivated us to conduct a numerical study of the effect of the aspect ratio on the main characteristics of the frequency-magnitude distributions generated by our model. With this in mind, we carried out a comparison between our results and those found by using different approaches. First, we define two equations that assign the number of cells according to the width $N_y$ and length $N_x$ of the domain.

$$N_x = (W_{\text{eff}}/L_{\text{eff}}) \times R_a \times L^*, \quad \text{and} \quad N_y = N_{\text{cell}}/N_x, \tag{10}$$

We define the aspect factor $R_a$ as a value that extends the $N_y$ side of the rectangle, assigning the number of cells in the width and length sizes of the domain, $L^* = \sqrt{N_{\text{cell}}}$. As $R_a$ increases the aspect ratio $\chi$ transforms the domain area into a thinner rectangle. Eq. 10 allows us to compare different aspect ratio values preserving the number of cells $N_{\text{cell}}$.

To perform this study, we chose as reference the fourth SA region because its width-length ratio is close to 1 (1.03), *i.e.,* making it a squared source ($L_{\text{eff}}/W_{\text{eff}} = 1$). In the experiment, we modified the ratio $R_a$ (as is observed in Fig. 10) in the algorithm, keeping constant the area in the computational domain, $N_{\text{cell}} = 10000$, as well as the other input parameters. In this work, we consider values of $R_a \geq 1$. The values of $\chi$ for the four seismic regions are shown in Table 2. It is worth mentioning that the aspect factor $R_a$ modifies the effective area and the asperity region in the same proportion.

In Fig. 10, we exemplify two different $R_a$ and their respective $\chi$ values, being (a) $R_a = 1$ and $\chi = 1.0$, (b) $R_a = 2$ and $\chi = 3.8$, considering the same number of cells ($N_{\text{cell}} = 10000$). The color bar indicates the strength value $\gamma$, with one corresponding to the minimum value assigned to the background area. The simulated asperity has a heterogeneous strength $\gamma_{\text{asp}}$, which is also larger than the background. We observe two main effects of the size variation of the computational domain on the frequency-magnitude curves:

1. The detected minimum magnitude. In our experiments, the effective source area (Table 1) remains constant, thus a finer mesh can support smaller ruptures, and therefore, TREMOL generates lower magnitude events.

2. The total number of triggered events, which is strongly dependent on the minimum magnitude observed in experiments.

However, large-magnitude behaviors are not affected by the increase or decrease of the computational mesh. In Fig. A1, we observe an example of frequency-magnitude distribution as function on the mesh size and the aspect-ratio, $R_a$.

### 6.1 Results of the Aspect Ratio influence

The aim of this section is to explore the effect of the fault aspect-ratio $R_a$ on the synthetic (frequency-magnitude) seismicity generated by TREMOL. In Fig. 11, we plot magnitude histograms for six different aspect ratios: Fig. 11 (a) $R_a = 1$, Fig. 11

(b) $R_a = 1.4$, Fig. 11 (c) $R_a = 1.7$, Fig. 11 (d) $R_a = 2.0$, Fig. 11 (e) $R_a = 2.1$, and Fig. 11 (f) $R_a = 2.4$. Fig. 11 illustrates the strong dependency of these magnitude distributions to the $\chi$ value. In our model, critical point is reaches for values of $R_a > 2.0$ ($\chi > 3.8$), at whose value TREMOL generates only a few events of large magnitude.

The behavior of the synthetic seismicity displayed in Fig. 11 is very interesting and shows a possible relation of the area size and shape in the transition between a GR distribution-type behavior and a characteristic-type. In the numerical experiments, we observe that narrow synthetic faults (large $R_a$ values, Figs. 11 and A1) produce large earthquakes and few low-magnitude events. The extreme behavior is observed for $R_a = 2.4$ where low-magnitude events disappear, and only one maximum magnitude event is generated. A possible explanation of this behavior could be related to the physical process observed in real scenarios, as analyzed by previous works (see Introduction references). For example, the conclusions in Wesnousky et al. (1983) offer an explanation for the observed numerical results because, in our model, the characteristic event is closely related to the fault length. Moreover, Sibson (1989) proposed that the seismogenic structures may have an influence on characteristic earthquakes. In TREMOL, the seismogenic structures are defined by the computational domain including its boundary conditions. The model boundaries are absorbent, i.e., the cells at the border dissipate a fraction of its load and no ruptures occur outside the edges. Therefore, TREMOL considers an inner seismogenic domain and an aseismic contour. As $R_a$ increases, the width of the seismogenic zone decreases and the fault rupture grows in length (Leonard, 2010). Moreover, as $R_a$ increases, the quantity of load that dissipates through the boundary increases because a larger number of cells lay in the frontier. Consequently, in the model, the seismicity distribution is clearly related to the aspect ratio of the simulated seismogenic region. As $R_a$ increases the system reduces the generation of a wide range of magnitude values, until it reaches a critical $R_a$ value $R_a \approx 2$ ($\chi \approx 4$), where the system is only able to generate very few but large earthquakes. From these results we observe that the TREMOL seismicity is highly sensible to the aspect-ratio, so tuning this parameter we can obtain either a GR-type or characteristic-type distribution. Larger $R_a$ values make it more likely for load to be dissipated at the boundary. With less energy available, secondary ruptures, either large or small, are thus inhibited.

In our results, we observed that the maximum magnitude is approximately 7.4, independently on the aspect-ratio. Nevertheless, as is seen in Fig. A1 the frequency-magnitude curve is clearly dependent on the aspect-ratio. Therefore, we pointed out that the maximum magnitude remains constant for all $R_a$ variations (Fig. 11). In that sense, we observed that the maximum magnitude is related to the asperity area and not to the aspect-ratio of the computational domain. As seen in our simulations the lack of low-magnitude events strongly depends on the aspect-ratio. However, in our results, the four mainshocks analyzed in this work (Table 1) have $\chi$ values between 1 and 2, and therefore, their frequency-magnitude histograms are similar to Fig. 11 (a) and (b). Those values agree with the results found in the referred previous works.

## 7   Discussion

The simulated TREMOL seismicity distributions show a high similarity to real seismicity curves associated with the four SUB3 reference mainshocks, for magnitude values of $M_w \geq 4$, if a proper scaling ~~magnitude-area~~ relation is adopted (see Figs. 4, 5, 6, 7, and Fig. 8). Moreover, we compare two different methodologies to obtain frequency-magnitude curves: the magnitude-area

relations described in Eqs. 5, 3, 4, and 6; and the moment-magnitude equation proposed in Kanamori and Anderson (1975). The moment is computed using Eq. 8, and requires the stress drop values. One conclusion is that seismicity distributions highly depend on the stress drop values when this approach is followed. Also, the moment magnitude relations always underestimated the maximum magnitudes, although low-magnitudes distributions were matched reasonably well (Figs. 4 (f), 5 (f), 6 (e), 7 (f).

In three of the study cases the best relationship is the one proposed by Ramírez-Gaytán et al. (2014), which leads to an excellent fit in the referred magnitude range (see Figs. 4, 5, and 7). Alternatively, there is one case where a better fitting is achieved by using the relation proposed by Somerville et al. (1999b) ( see Fig. 6). Finally, the synthetic aggregated curves in Fig. 8 shows that the scaling of Ramírez-Gaytán et al. (2014) in Eq. 6 allows a better global fit when considering the four SA regions in SUB3. This relation was developed using $M_w \in$ [6.9-8.1] earthquakes in the Mexican subduction zone, hence it seems reasonable that this relation works well for the main events studied in this paper.

It is worth pointing out the cases including events of magnitude lower than $M_w = 4$, where synthetic curves usually overestimate the real seismicity curves (Figs. 4, 5, 6, 7, and 11). This may occur because the number of events in the seismic catalog is not enough to compare with the synthetic ones due to the limitations of the network. In support of this assumption, we emphasize the results for the 2012 mainshock that has the largest number of associated events, since it was for this case that TREMOL was able to more closely match the observed distribution even for small magnitudes of $M_w \geq 3$.

In summary, we can conclude that the synthetic seismicity distributions agree well with the observations related to the four earthquakes of magnitude $M_w > 4$ used as study cases. The good agreement achieved by the synthetic frequency-magnitude curves, support the assumption in Zúñiga et al. (2017) that attribute this type of distribution to ruptures of single asperities and provide further support to the hypothesis that regions where ruptures are simple, yield relations that depart from the linearity of the common GR law, indicating a process of characteristic events. Moreover, as a way to provide an additional counterexample of the capabilities of TREMOL in the case of absence of a hard asperity area, we modeled the expected seismicity on the SA region 4 under uniform $\pi$ and $\gamma$ values. We allow for these conditions by taking $\pi_{\mathrm{asp}} = \pi_{\mathrm{bkg}} = 0.67$, and $\gamma_{\mathrm{asp}} = \gamma_{\mathrm{bkg}} = 1$. Fig. 12 compares this new synthetic frequency-magnitude curve, in the absence of a hard asperity, with the previous TREMOL result that accounts for a single asperity (and shown in Fig. 7), and the real seismicity. Fig. 12 proves that the asperity condition in TREMOL is indeed a mandatory requirement to reproduce the seismicity features observed in the SUB3 region.

Lastly, TREMOL results in Fig. 11 reveal a frequency-magnitude distribution sensitivity to the fault aspect-ratio $\chi$. As $\chi$ increases, *i.e.,* the effective area is modeled as a long-rectangle, the synthetic frequency-magnitude distribution changes until the critical $\chi \approx 4$ value is reached, above which only large magnitude events are triggered. These results indicate that the shape of the model domain controls the frequency-magnitude distribution of the synthetic data. In general, square fault areas allow the generation of a large variety of magnitude events. On the other hand, as the asperity area becomes a long rectangle, TREMOL generates only a few large events. Thus, in our model, we can control the transition of the magnitude distribution through the $\chi$ parameter. Moreover, our numerical results agree with observational studies that find a similar $\chi \approx 2$ for dip-slip faulting style.

# 8 Conclusions

The frequency-magnitude distribution has a significant impact on the seismic hazard assessment. Asperities seem to have a direct relation with the occurrence of preferred size events. In this work, we set to demonstrate the capability of the model employed in TREMOL to generate seismicity distributions similar to those observed in region SUB3 of the subduction regime of Mexico for magnitudes $M_w > 4$. Our simulation results support the hypothesis presented by Singh and Mortera (1991) and Zúñiga et al. (2017) that ascribe mainshocks occurring in region SUB3 as mainly generated from rupture of single asperities (Fig. 12). Nevertheless, we also find an impact on the synthetic curves that depends on the area-magnitude scale relation. We find that in four cases out of five, the relation that better fits the synthetic in relation to the real curve is that proposed by Ramírez-Gaytán et al. (2014). It is worth to note that this relationship was developed for earthquakes from the Mexican subduction zone, hence it is expected to work well to describe the magnitude-area relation of the events simulated in this paper.

TREMOL makes it possible to analyze regions where seismic data is too limited. In this sense, is highlighted that we use as input data the information of four large earthquakes, but the number of events generated approaches 1000. Furthermore, we find that our model agrees with the results obtained in other studies that emphasize the importance of the fault aspect ratio $\chi$ on the frequency-magnitude distribution. Nevertheless, results for the four analyzed sub-seismic regions indicate that the behavior of the synthetic histograms matches well the observed ones in the range $1 \leq \chi < 4$.

Results further encourage us to continue exploring the capabilities of our model, for future applications of TREMOL for the modeling of seismicity distributions at other subduction zones, such as Chile or Japan. In addition, we continue working on more general rupture models by including tridimensional fault systems, sources interactions such as that produced in the doublets phenomena, as well as a reloading process that allows the generation of the seismic cycle.

*Code availability.* The TREMOL code is freely available at GitHub repository (https://github.com/monterrubio-velasco), or by requesting the author (marisol.monterrubio@bsc.es). In all cases, the code is supplied in a manner to ease the immediate execution under Linux platforms. Preprocessing, run control and postprocessing scripts covering every data processing action for all the results reported in this work is provided https://github.com/monterrubio-velasco/TREMOL_singlets/tree/TREMOL_singlets_SUB3study. User's manual documentation are provided in the archive as well.

*Data availability.* Data sets are available through Monterrubio-Velasco et al. (2019a), Rodríguez-Pérez et al. (2018), and SSN (2019).

*Author contributions.* MMV developed TREMOL v0.1.0 code and the methodology used in this paper. MMV, RZ, AAM, OR, QRP and JP provided guidance and theoretical advice during the study. All the authors contributed to the analysis and interpretation of the results. All the authors contributed to the writing and editing of the paper.

*Competing interests.* The authors declare that they have no conflict of interest.

*Acknowledgements.* We thank the two anonymous reviewers whose comments and suggestions helped improve and clarify this manuscript. The research leading to these results has received funding from the European Union's Horizon 2020 research and innovation programme under the grant agreement Nº 823844, ChEESE CoE Project (last access: 6 November 2020). M. Monterrubio-Velasco, O. Rojas and J. de la Puente thanks to ChEESE CoE Project. Quetzalcoatl Rodríguez-Pérez was supported by the Mexican National Council for Science and Technology (CONACYT) (Cátedras program- project 1126).

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

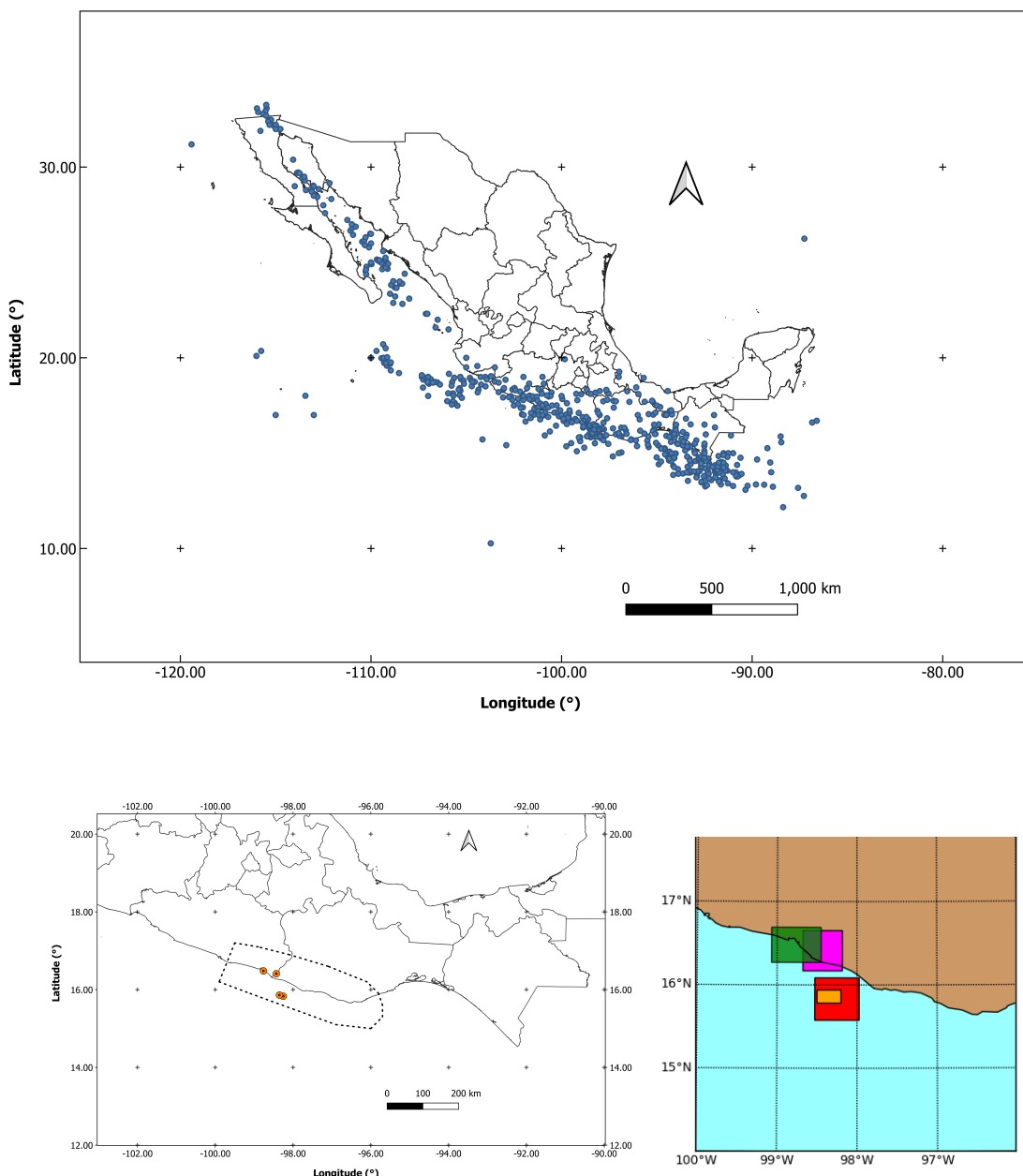

**Figure 1.** a) Map of earthquake epicenters in Mexico. From 1900 to 1973, events have magnitudes greater or equal to 6.5, and from 1974 to June 30, 2019, events have magnitudes greater or equal to 5.5 (SSN, 2019). b) Map of epicenters (orange circles) of the four earthquakes described in Table 1. The SUB3 region is the polygon enclosed by the dotted line. c) Effective areas of the four earthquakes of Table 1.

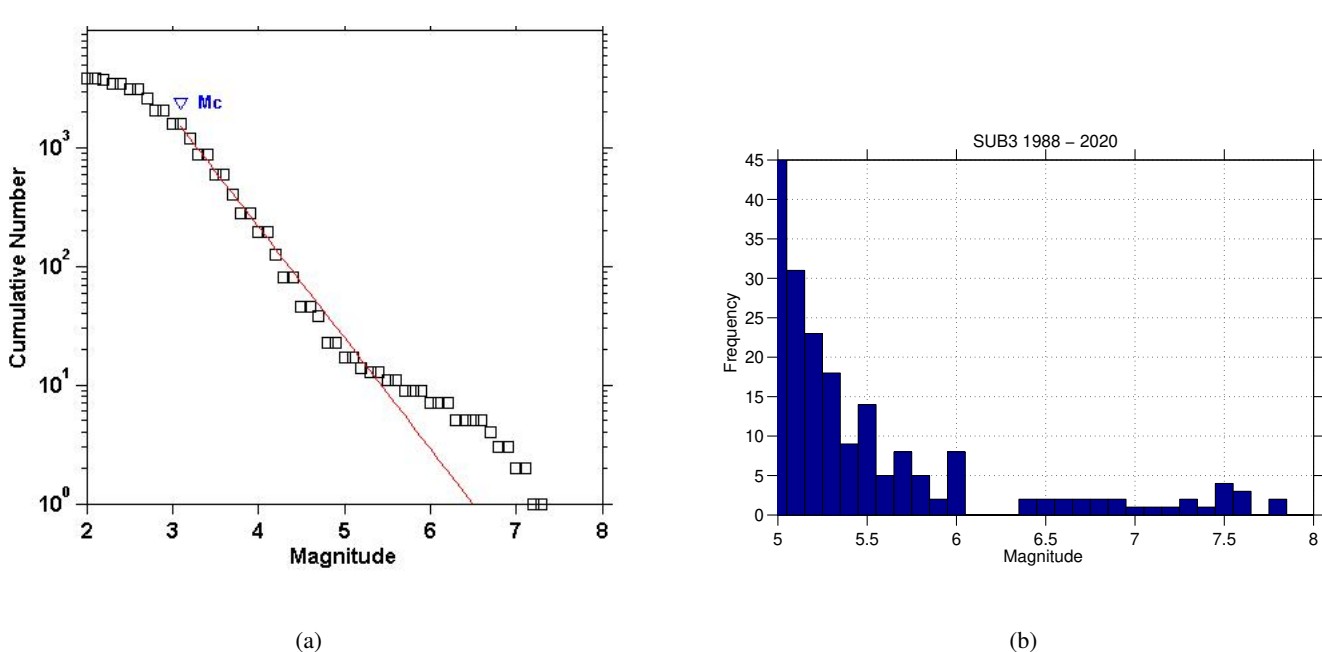

(a)

(b)

**Figure 2.** (a) Frequency-magnitude distribution of events occurred in the *SUB3* seismic region after 1988-2014 (Zúñiga et al., 2017). (b) Magnitude histogram for M>5.0 from 1988 to 2020.

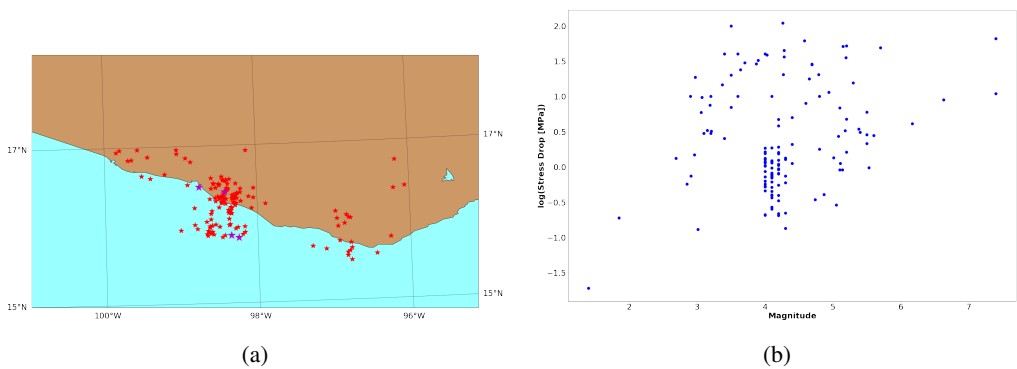

(a)                                                                        (b)

**Figure 3.** (a) Spatial distribution of earthquakes with computed stress drop in Rodríguez-Pérez et al. (2018) and the four analyzed earthquakes in red and magenta stars, respectively (b) Stress drop *vs* magnitude distribution of red stars events.

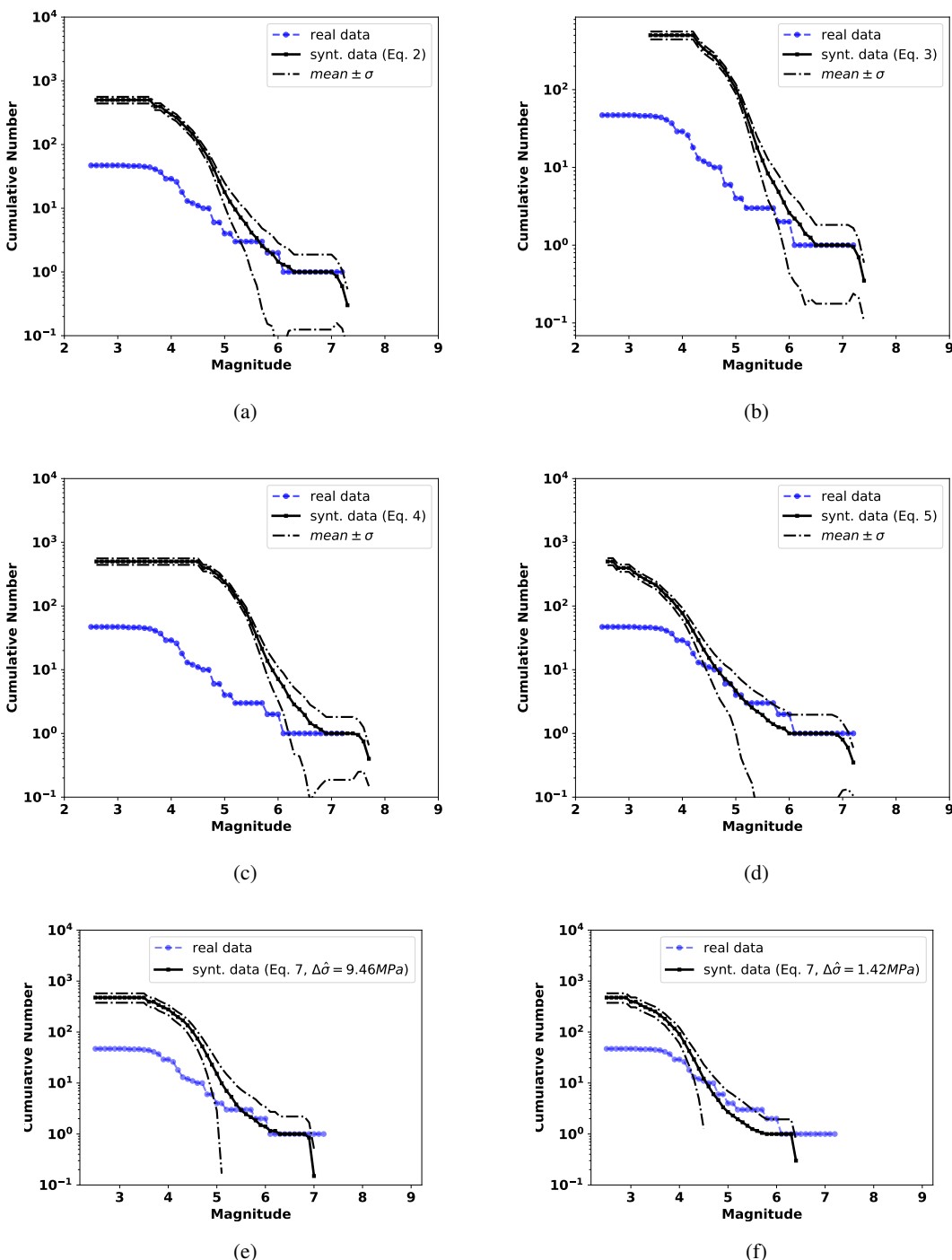

**Figure 4.** Observed and TREMOL synthetic frequency-magnitude curves for SA region 1 (Table 1). The solid black line indicates the mean after 20 realizations and the broken lines the standard deviation. The blue line is the observed seismicity distribution for events from 1988 to half year before the mainshock date (14/09/1995), including events occurred at the depth range mentioned in the text. Each mean curve is obtained by the application of one of the four magnitude-area relations used in this work: (a) Eq. 3, (b) Eq. 4, (c) Eq. 5 , (d) Eq. 6, (e) and (f) Eq. 7 using the mean and median stress drop values, respectively.

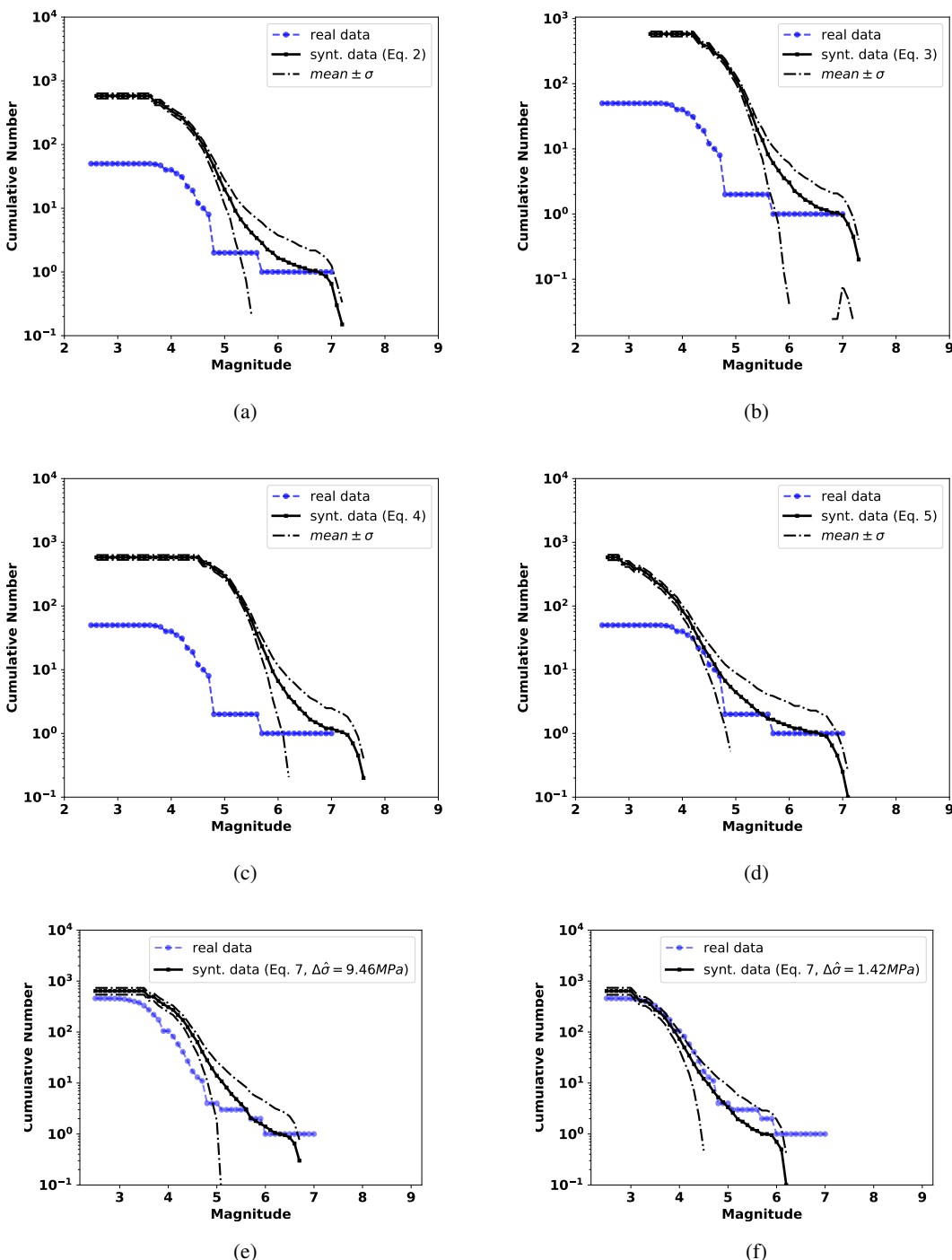

**Figure 5.** Observed and TREMOL synthetic frequency-magnitude curves for SA region 2 (Table 1). The solid black line indicates the mean after 20 realizations and the broken lines the standard deviation. The blue line is the seismicity curve for events from 1988 to half year before the mainshock date (25/02/1996), including earthquakes occurred at the depth range mentioned in the text. Other features as in Fig. 4

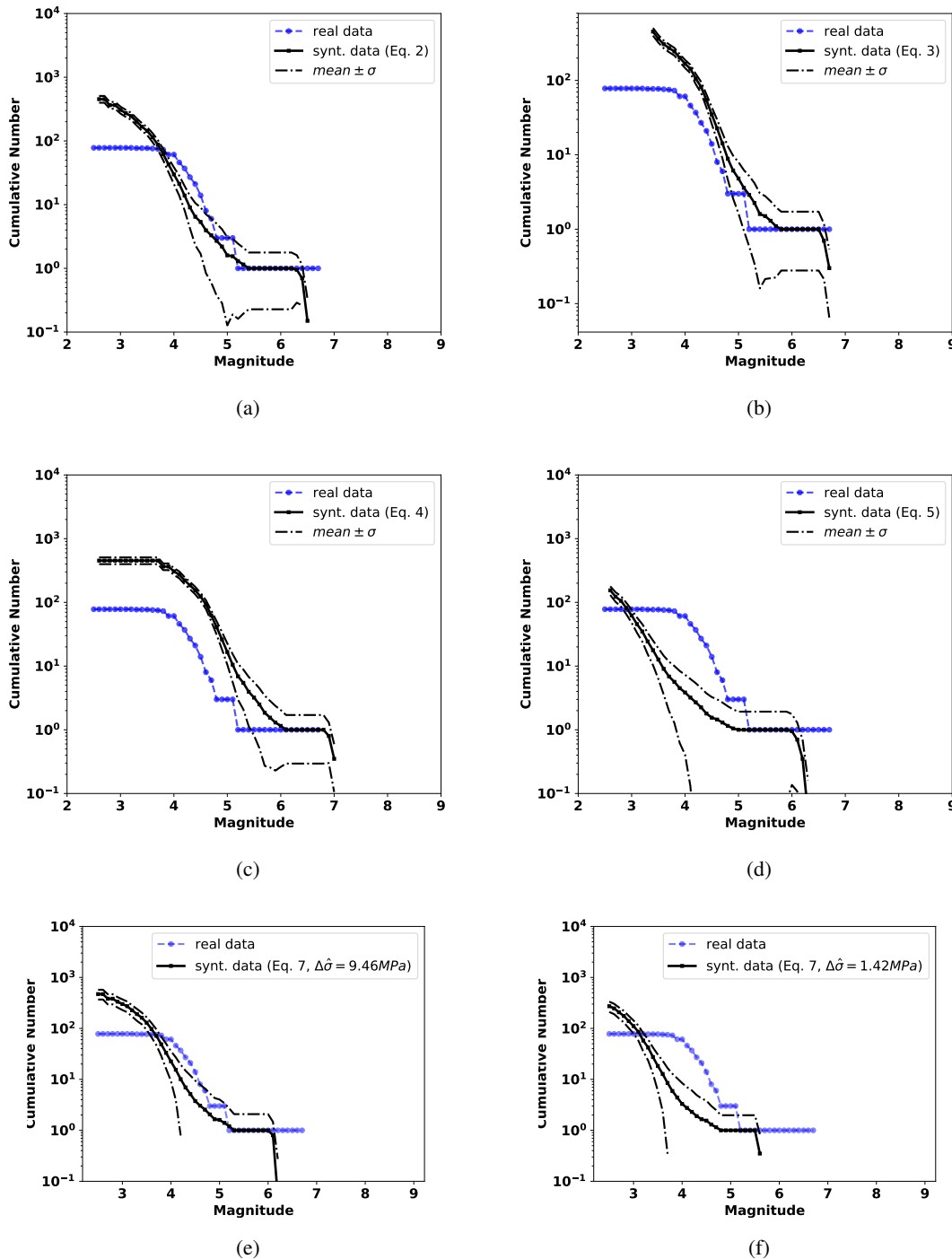

**Figure 6.** Observed and TREMOL synthetic frequency-magnitude curves for SA region 3 (Table 1). The solid black line indicates the mean after 20 realizations and the broken lines the standard deviation. The blue line is the seismicity curve for events from 1988 to half year before the mainshock date (19/07/1997) including earthquakes occurred at the depth range mentioned in the text. Other features as in Fig. 4

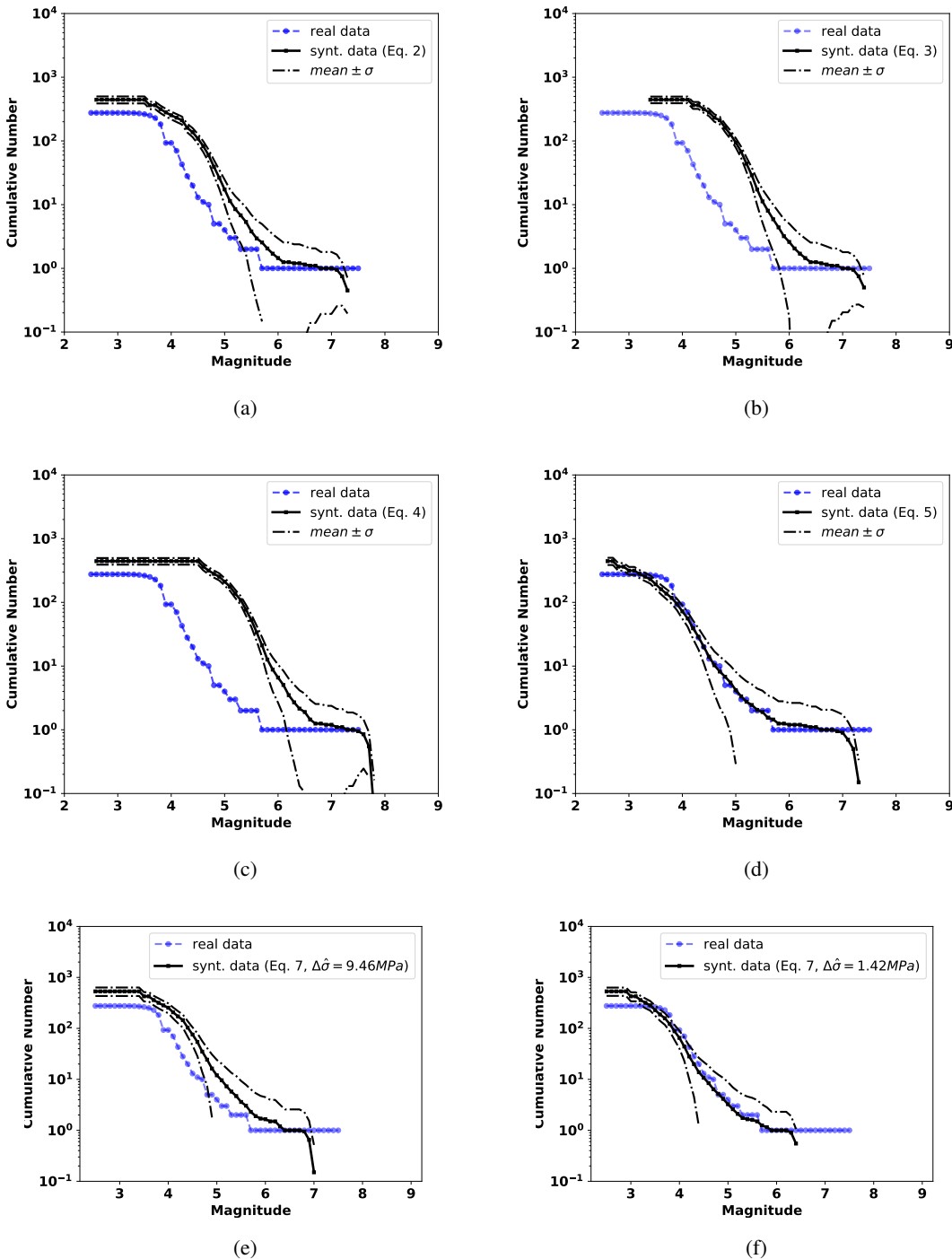

**Figure 7.** Observed and TREMOL synthetic frequency-magnitude curves for SA region 4 (Table 1). The solid black line indicates the mean after 20 realizations and the broken lines the standard deviation. The blue line is the seismicity curve for events from 1988 to half year before the mainshock date (20/03/2012) including earthquakes occurred at the depth range mentioned in the text. Other features as in Fig. 4

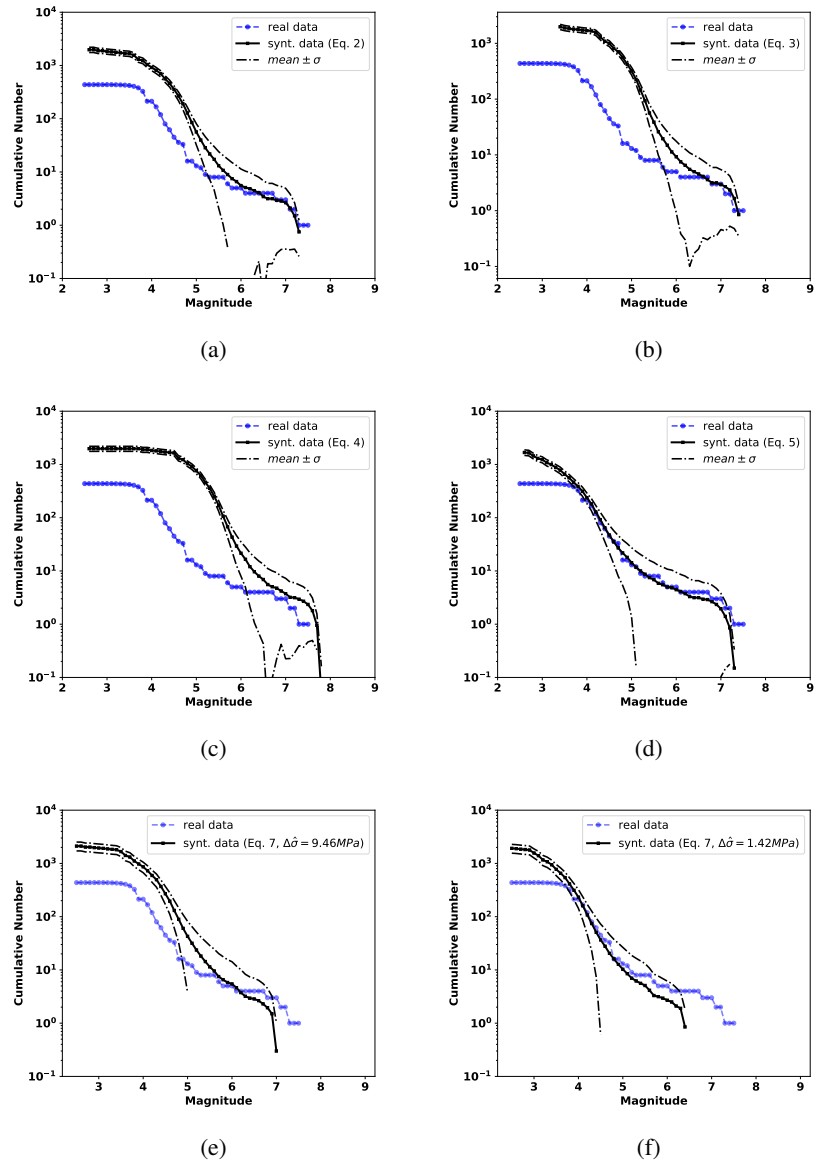

**Figure 8.** Observed and TREMOL synthetic frequency-magnitude curves for the aggregated frequency-magnitude curves computed with the contribution of the four mainshock ruptures. The solid black line is the mean of the synthetic results considering 80 realizations and the broken lines the standard deviation. The blue line is the seismicity curve for events of Figs. 4, 5, 6, and 7. Each mean curve is obtained from one of the four magnitude-area relationships used in this study: (a) Eq. 3, (b) Eq. 4, (c) Eq. 5, (d) Eq. 6, (e) and (f) Eq. 7 using the mean and median stress drop values, respectively.

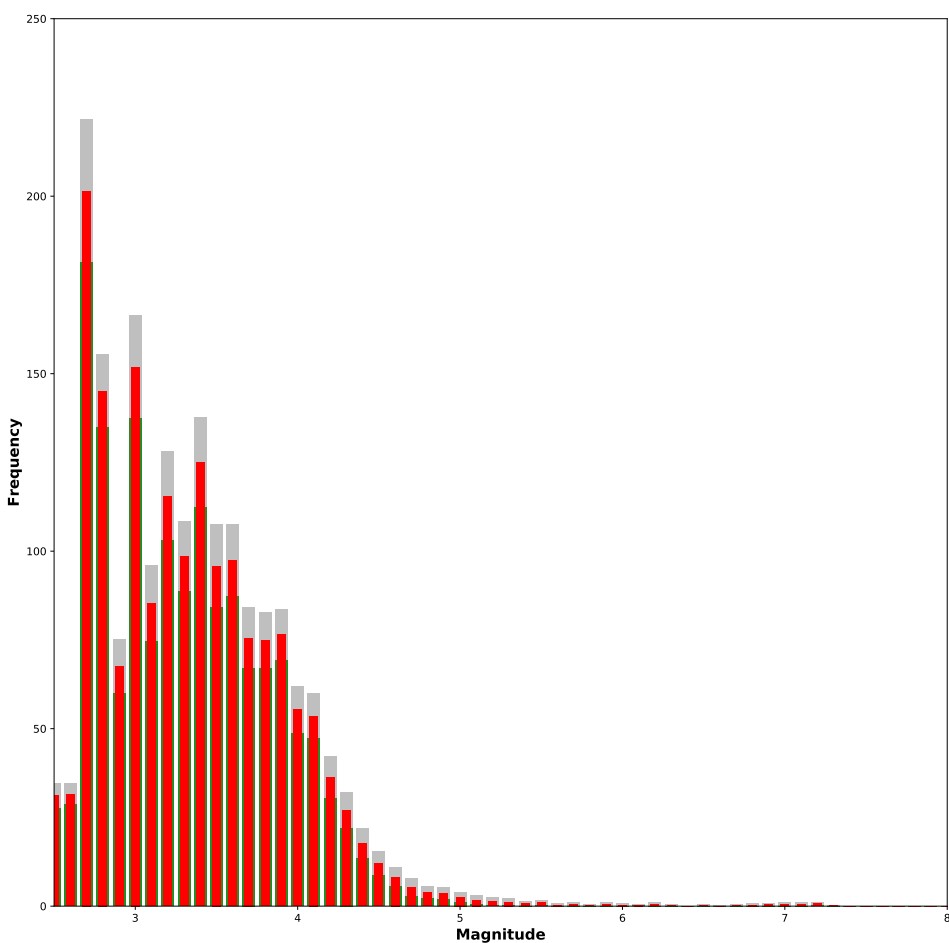

**Figure 9.** Magnitude histogram computed from the aggregated synthetic seismicity using Ramírez-Gaytán et al. (2014) relation. Gray and green bars indicates the standard deviation from the statistical analysis. Blue bars the sum of the mean values per magnitude intervals.

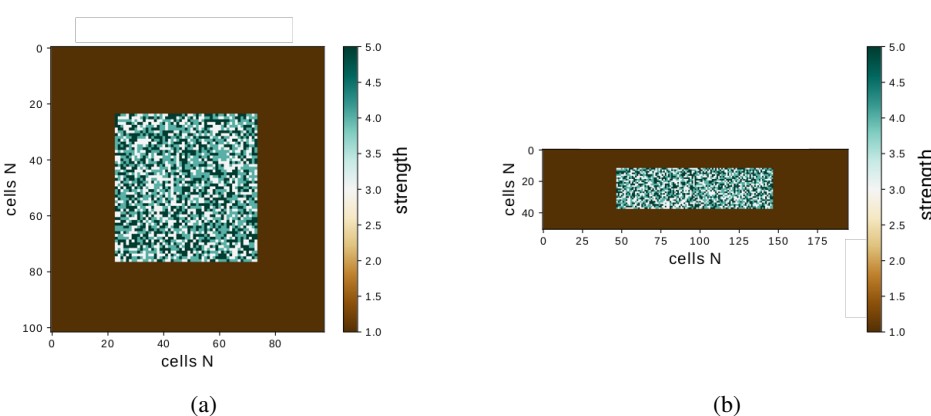

**Figure 10.** Example of two sub-seismic regions with different aspect ratio, $\chi = L_{\mathrm{eff}}/W_{\mathrm{eff}}$. The number of cells remains constant in both cases $N_{\mathrm{cell}} = 10000$. (a)$R_{\mathrm{a}} = 1$ and $\chi = 1.0$, (b) $R_{\mathrm{a}} = 2$ and $\chi = 3.8$.

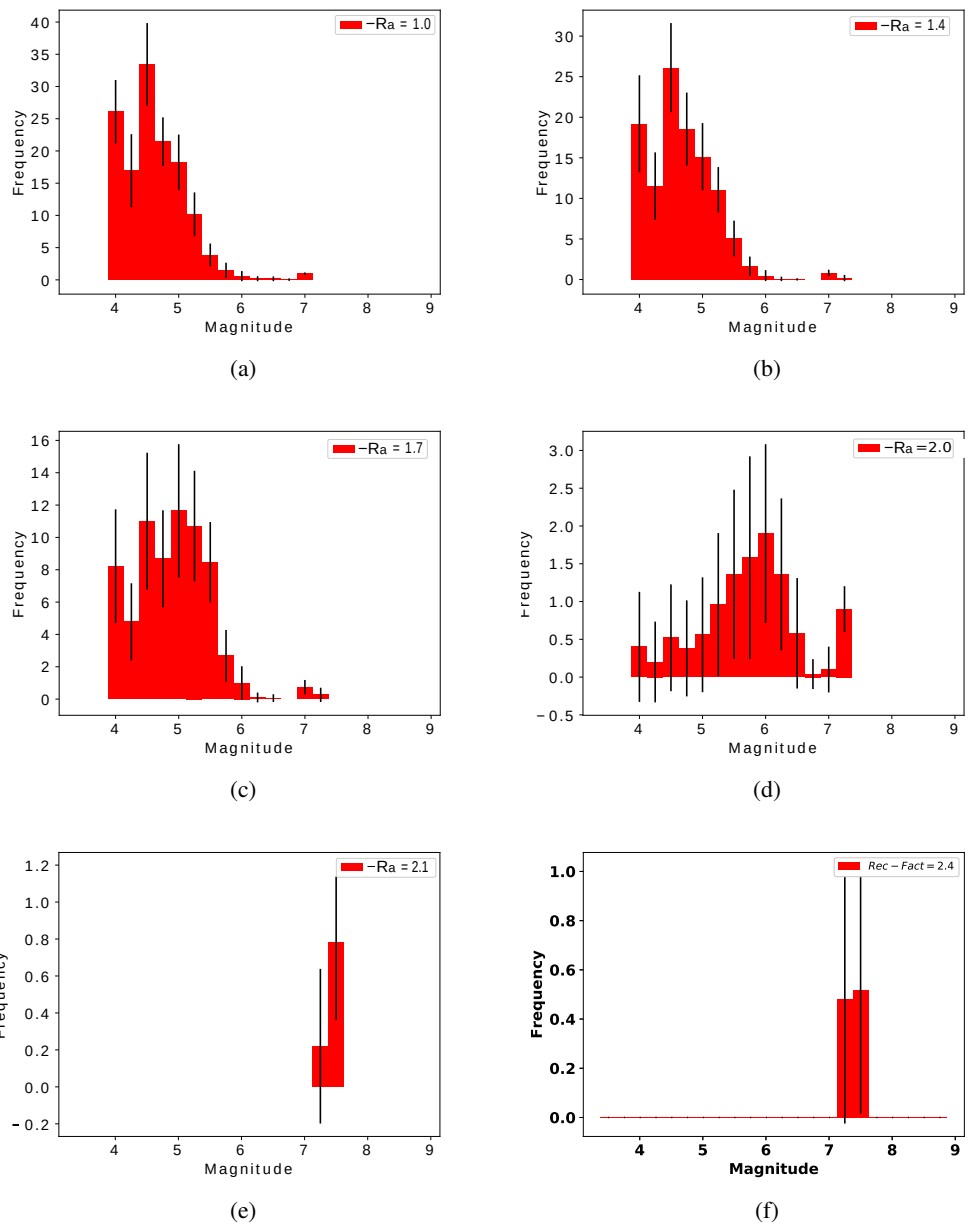

**Figure 11.** Frequency-magnitude histograms as function of the ratio size $R_a$ and the area domain of 100000 cells. The bars shows the mean histogram, and the error bars depicts the standard deviation of the twenty realizations.

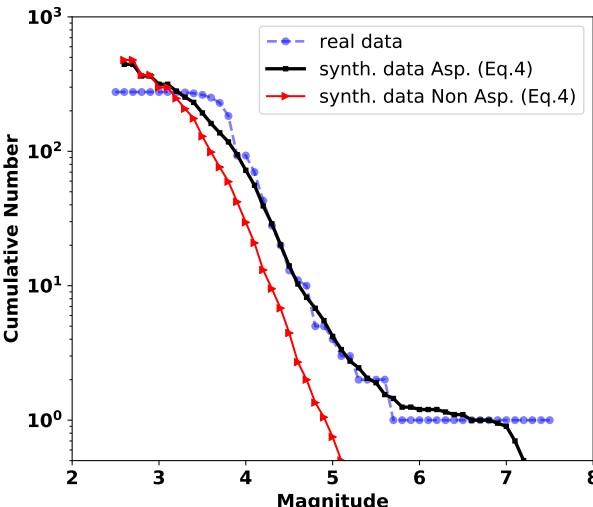

**Figure 12.** The real and TREMOL frequency-magnitude curves for the SA region 4. The solid black line represents the mean synthetic seismicity curve of 20 realizations considering one single asperity in the domain. The red line corresponds to the mean synthetic seismicity curve of 20 realizations without any single asperity in the domain.

**Table 1.** Data of four large earthquakes occurred in SUB3 and reported by Rodríguez-Pérez et al. (2018). Specifically, the date of occurrence; moment magnitude $M_{\mathrm{w}}$; effective length $L_{\mathrm{eff}}$ and wide $W_{\mathrm{eff}}$ of the area following the methodology in Mai and Beroza (2000); asperity area $A_{\mathrm{a}}$; and the epicentral coordinates $Lon$ and $Lat$.

| Date | $M_{\mathrm{w}}$ | $L_{\mathrm{eff}}$ $[km]$ | $W_{\mathrm{eff}}$ $[km]$ | $Z$ $[km]$ | $S_{\mathrm{a}} =$ $A_{\mathrm{a}}/A_{\mathrm{eff}}$ | $Lon$ Deg. | $Lat$ Deg. | color* | SA region |
|------|------|------|------|------|------|------|------|------|------|
| 14/09/1995 | 7.4 | 68.80 | 46.61 | 16 | 0.23 | -98.76 | 16.48 | green | 1 |
| 25/02/1996 | 7.1 | 61.70 | 56.54 | 25 | 0.18 | -98.25 | 15.83 | red | 2 |
| 19/07/1997 | 6.5 | 23.27 | 17.51 | 15 | 0.26 | -98.35 | 15.86 | orange | 3 |
| 20/03/2012 | 7.4 | 54.94 | 53.59 | 19 | 0.26 | -98.43 | 16.41 | magenta | 4 |

**Table 2.** Additional data of the sub-seismic regions: area and aspect ratio.

| sub-sesimic region | $Area$ $[km^2]$ | Aspect ratio $\chi$ | color* |
|---|---|---|---|
| 1 | 3206.77 | 1.48 | green |
| 2 | 3488.52 | 1.09 | red |
| 3 | 407.46 | 1.33 | orange |
| 4 | 2944.23 | 1.03 | magenta |

*color area in-Fig. 1 (c).

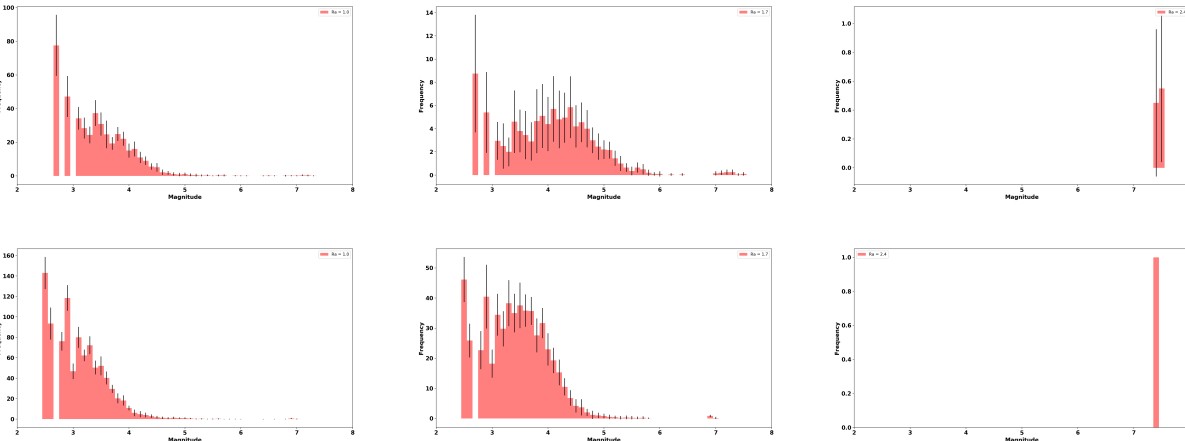

**Figure A1.** Magnitude histograms as function of the ratio size $R_a$ (from left to right $R_a = 1.0, 1.7, 2.4$, respectively), and the effective area 40000 cells and 90000 cells, for upper and lower row figures respectively. The bars shows the mean histogram, and the error bars depicts the standard deviation of the twenty realizations.

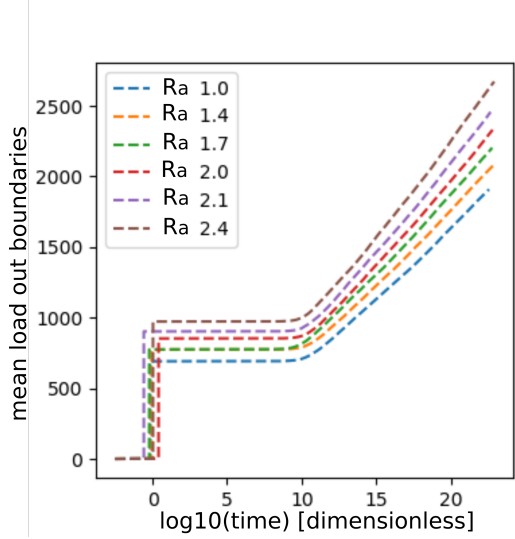

**Figure A2.** Evolution of the mean load dissipated from the system through the border as a function of $R_a$.

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
