# Peer review of "Synthetic seismicity distribution in Guerrero-Oaxaca subduction zone, México and its implications on the role of asperities in Gutenberg-Richter Law"

_Geoscientific Model Development, 2020_

## Referee Comment (RC1) · Anonymous Referee #1 · 11 Aug 2020

The scaling relation of seismicity frequency-magnitude is an important topic for the application of seismic hazard assessment. This paper investigates the possible role of asperities in the seismicity frequency-magnitude relations by using their developed code, TREMOL v0.1.0. This paper also studies how the aspect ratio of asperities affects the seismicity frequency-magnitude scaling relations. This is a very interesting idea. But, I find in this paper the conclusion based on the current tests are not sufficiently convincing. For instance, the magnitude shall not only depend on the ruptured area but also depends on the stress drop or the final slip. In the current version, the

effects of the aspect ratio on the final slip and magnitude are ignored somehow. In addition, in Section 6, the number of cells in the computational domain might affect the seismicity frequency-magnitude curve, which was fixed. Therefore, I recommend a major revision for this paper, with my suggestions listed below.

In another paper of theirs, which introduces the code TREMOL, I find they considered the stress drop of each broken patches. Combining the rupture area and the stress drop, they can uniquely determine the magnitude of each earthquake, such as using the inversion of Okada's matrices. This is important because, given the same rupture area and stress drop, the magnitude of earthquakes also depends on the aspect ratio [Leonard, 2010; Hanks and Bakun, 2002]. So, I suggest the authors estimating the magnitude based on the numerical methods, rather than the empirical magnitude-area relations (equations 2-5).

Line 170: They consider each SA region as independent for an individual TREMOL simulation. But these four regions can affect each other by the static stress perturbation, such as the Coulomb stress. In addition, each asperity may have different earthquake cycles due to various loading condition and their TREMOL implementation does not allow simulating a full earthquake cycle. It might be tricky to simply combine all SA curves into one synthetic aggregated curve. At lease, the authors shall discuss the possible effects of this procedure in the manuscript.

Fig. 12 is very interesting but hard to understand. Do these two models have the same effective width? Why does the narrow fault tend to produce larger earthquakes? Based on the fracture mechanics theory, wider faults (larger elastic energy release) are more likely to propagate larger earthquakes. More explanations for this figure are needed.

Line 275 – "In that sense, we could conclude that the maximum magnitude is related to the total rupture area and not to its aspect ratio or shape". This is not correct if the aspect ratio is large. Magnitude depends on final slip. Given the same stress drop, the final slip depends on the shorter dimension of the rupture areas if the aspect ratios

are high. From the observations, the scaling relation between magnitude and rupture area is different for aspect ratio =1 and >1 (See the difference between the L-model and W-model [Hanks and Bakun, 2002]).

References:

Leonard, M. (2010). "Earthquake fault scaling: self-consistent relating of rupture length, width, average displacement, and moment release." Bulletin of the Seismological Society of America 100(5A): 1971-1988.

Hanks, T. C. and W. H. Bakun (2002). "A bilinear source-scaling model for M-log A observations of continental earthquakes." Bulletin of the Seismological Society of America 92(5): 1841-1846.
* * *

---

## Referee Comment (RC2) · Anonymous Referee #2 · 1 Sep 2020

In the manuscript the authors conducted synthetic seismicity distribution by considering the aspect ratio of a single asperity, and compared the numerical results with observations in the Mexican subduction zone. In general, the synthetic results and data showed good consistency. The authors also pointed out that the aspect ratio of the asperity played critical roles in magnitude-frequency distribution. Over a critical value the system tended to become dominated by characteristic earthquakes. I think that this study is important to evaluate seismic hazard. But the manuscript needs substantial improvement, mostly in structure and writing. Below please find my detailed

**comments.**

The motivation in the Introduction was not clear. It is pretty odd to have such a long paragraph, with inserting bullet points on specific regional settings. It is very difficult to get the major points of this study, and what might be innovative comparing with previous approaches that had been carried out in the same region. I feel that rewriting the Introduction is necessary, and recommend the following structure based on my reading.

You may start from the significant of G-R law on probabilistic seismic hazard analysis, but it is necessary to highlight why synthetic seismicity distribution is important (the major point to deliver in this study). For a region that has very few earthquakes, it is straightforward to produce synthetic seismic distribution. But for the Mexican subduction zone where earthquakes are frequent, with recent great earthquakes (M>8), why is it significant and necessary to conduct such synthetic GR law analysis?

Then you can highlight the important of asperities and the effects of aspect ratios of asperities on GR law. If it has not been considered in previous synthetic seismicity generator, it serves as a natural innovative point of this study.

It will be very helpful to explain how the asperities were identified in different subduction zones that were mentioned in lines 35-40. Why do they have anything to do with TREMOL, which was first introduced in line 41? The statement here was really disjointed and should be rewritten.

Overall the grammar in the manuscript is OK, but there are lots of fragmented and/or long sentences, which are sometimes awkward to read. For example, the first sentence in section 2 contains "low magnitude previous events", which should be "previous low-magnitude events". Many other words share the similar problems and need a thorough editing work.

Some texts/paragraphs should belong to Introduction or Discussion, but were mixed in
different sections. Here I listed a few, but not all of them. Line 59-61: statement of asperities should be moved into Introduction. Some descriptions in section 3 could be moved into Introduction as well.

For the results, I think the implications of Figure 12 are important on earthquake physics. What are the potential underlying physics that lead to such transition of GR distribution to a more characteristic earthquake pattern?

Figure 2-4 can be either merged together, or grouped together to better illustrate the locations of observed seismicity.

I suggest deleting "for any type of" in the first sentence of the abstract.

---

## Author Comment (AC1) · 7 Oct 2020

Dear reviewer,

First of all, thanks a lot for your valuable time and comments that help us to improve the manuscript. We have considerably revised the paper following your recommendations. In the next pages, we answer all your questions. We have adopted the following format in our answers:

Question/comment from the reviewer (in bold) Lines in the manuscript where the answer is addressed (–>) Answers or replies from the authors (no special format) New paragraphs added to the manuscript (in italic font)

We are at your disposal to provide any further information you may request, and well satisfied after adding to our manuscript all the new plots, figures, and bibliography files, that are detailed in this reply.

Kind regards, Marisol Monterrubio-Velasco and coauthors.
* * *
1. The magnitude shall not only depend on the ruptured area but also depends on the stress drop or the final slip.

1a) In the current version, the effects of the aspect ratio on the final slip and magnitude are ignored somehow.

→ line 197 - 199

TREMOL is capable of estimating the rupture areas assigning physical units to the numerical domain. In this paper, we do not consider slip to compute the magnitude distributions. On the other hand, TREMOL is not able to model the stress drop since the tectonic load is simulated using dimensionless units. We estimate a mean load drop, not related to any physical unit.
* * *
1b) In addition, in Section 6, the number of cells in the computational domain might affect the seismicity frequency-magnitude curve, which was fixed.

→ lines 348 - 354

In order to answer this question, we carried out new simulations where we increase the area of the computational domain. In Fig. A1, we include magnitude histograms for three different Ra values to show the behavior of the frequency-magnitude as a func-

tion on area domain size. Based on our conclusions, we added the next explanatory paragraph,

"We observe two main effects of the size variation of the computational domain on the frequency-magnitude curves: 1. The observed minimum magnitude. In our experiments, the effective source area (Table 1) remains constant, thus a finer mesh can support smaller ruptures, and therefore, TREMOL generates lower magnitude events. 2. The total number of triggered events, which is strongly dependent on the minimum magnitude observed in experiments. However, large-magnitude behaviors are not affected by the increase or decrease of the computational mesh. In Fig. A1, we observe an example of frequency-magnitude distribution as function on the mesh size and the aspect-ratio, Ra"
* * *
2. In another paper of theirs, which introduces the code TREMOL, I find they considered the stress drop of each broken patch. Combining the rupture area and the stress drop, they can uniquely determine the magnitude of each earthquake, such as using the inversion of Okada's matrices. This is important because, given the same rupture area and stress drop, the magnitude of earthquakes also depends on the aspect ratio [Leonard, 2010; Hanks and Bakun, 2002]. So, I suggest the authors estimate the magnitude based on the numerical methods, rather than the empirical magnitude-area relations (equations 2-5).

→ lines 186 - 195

In order to compare the magnitude-area relations to other magnitude estimations, we use the magnitude-moment equation provided in Leonard (2010). We also include a new figure (Fig. 3) to show the spatial distribution of the stress drop database that we used to compute a mean and median stress drop value. And also we add a magnitude-stress drop plot to show the non correlation between this parameters
* * *
3. Line 170: They consider each SA region as independent for an individual TREMOL simulation. But these four regions can affect each other by the static stress perturbation, such as the Coulomb stress.

→ lines 234 - 237

We agree with the referee. Each SA region is modeled as independent and individual sources, and we are not considering any interaction between them. However, future TREMOL versions pretend to introduce the interaction between different asperities regions. In our model, the Coulomb stress change is simulated by the load transfer between the ruptured cells to its neighbors.

"Is worth mentioning that TREMOL 0.1.0 does not model the simultaneous interaction among the four sources, i.e., the Coulomb stress changes from one source to the next are not considered. However, the objective of this exercise is to aggregate the curve as an example of the aggregated seismicity without considering the interaction between sources. Future TREMOL generalizations would include such interactions."
* * *
4. In addition, each asperity may have different earthquake cycles due to various loading condition and their TREMOL implementation does not allow simulating a full earthquake cycle

→ lines 137-139

We already addressed this point in the revised manuscript.

"The current TREMOL implementation does not allow simulating a full earthquake cycle, because most of the tectonic load is spent during the whole process of the mainshock rupture and foreshocks, and no extra load is added during the simulation"
* * *
5. It might be tricky to simply combine all SA curves into one synthetic aggregated curve. At lease, the authors shall discuss the possible effects of this procedure in the manuscript.

As we mention above we are not considering the interaction between sources in this model version. However, in future versions we will incorporate this observed feature. In lines 234 - 237 we include a discussion on that.
* * *
6. Fig. 12 is very interesting but hard to understand.

→ lines 362 - 375

We add a paragraph including a possible explanation related to what we observe in the numerical results and the real seismicity behavior. Moreover, we add references of other previous works to support our conclusions of the results found in this figure, them included in the Introduction section. Moreover, we move some introductory phrases of Section 6 to the introduction to improve the reading. We also include a new figure (Fig. 1) to graphically illustrate the results analysis.

"The behavior of the synthetic seismicity displayed in Fig.11 is very interesting and shows a possible relation of the area size and shape in the transition between a GR distribution-type behavior and a characteristic-type. In the numerical experiments, we observe that narrow synthetic faults (large Ra values, Figs. 11 and A1) produce large earthquakes and few low-magnitude events. The extreme behavior is observed for Ra=2.4 where low-magnitude events disappear, and only one maximum magnitude event is generated. A possible explanation of this behavior could be related to the physical process observed in real scenarios, as analyzed by previous works (see Introduction references). For example, the conclusions in Wesnousky et al., (1983) offer an explanation for the observed numerical results because, in our model, the characteristic event is closely related to the fault length. Moreover, Sibson (1989) proposed that

the seismogenic structures relate to the characteristic earthquakes. In TREMOL, the seismogenic structures are defined by the computational domain including its boundary conditions. The model boundaries are absorbent, i.e., the cells at the border dissipate a fraction of its load and no ruptures occur outside the edges. Therefore, TREMOL considers an inner seismogenic domain and an aseismic contour. As Ra increases, the width of the seismogenic zone decreases and the fault rupture grows in length Leonard (2010). Moreover, as Ra increases, the quantity of load that dissipates through the boundary increases because a larger number of cells lay in the frontier (Fig. A2). Consequently, the quantity of energy inside the seismogenic zone is lower as Ra increases, and the system is only able to generate few but large earthquakes related to the asperity area." _______________________________________________________ 7. Do these two models have the same effective width?

line 349

No, the models have the same effective area but the width and length is modified following Eq. (8)
* * *
8. Why does the narrow fault tend to produce larger earthquakes?

We       include       a       possible       explanation       in       lines       362       -       375.
* * *
9. Based on the fracture mechanics theory, wider faults (larger leastic energy release) are more likely to propagate larger earthquakes. More explanations for this figure are needed.

We include a possible explanation in lines 362 - 375.
* * *
10. Line 275 – "In that sense, we could conclude that the maximum magnitude is

related to the total rupture area and not to its aspect ratio or shape". This is not correct if the aspect ratio is large. Magnitude depends on the final slip. Given the same stress drop, the final slip depends on the shorter dimension of the rupture areas if the aspect ratios are high. From the observations, the scaling relation between magnitude and rupture area is different for aspect ratio =1 and >1 (See the difference between the L-model and W-model [Hanks and Bakun, 2002]).

→ line 376 - 380

Our explanation was not complete, we clarified the comments including a sentence in the manuscript.

"In our results, we observed that the maximum magnitude is approximately 7.4, independently on the aspect-ratio. Nevertheless, as is seen in Fig. 2 the frequency-magnitude curve is clearly dependent on the aspect-ratio. Therefore, we pointed out that the maximum magnitude remains constant for all Ra variations (Fig. 11 in the manuscript). In that sense, we observed that the maximum magnitude is related to the asperity area and not to the aspect-ratio of the computational domain. As seen in our simulations the lack of low-magnitude events strongly depends on the aspect-ratio."

Please also note the supplement to this comment:
https://gmd.copernicus.org/preprints/gmd-2020-115/gmd-2020-115-AC1-supplement.pdf

―――――――――――――――――――――――

[Figure]

**Fig. 1.**

**Appendix A**

**A1**

[Figure]

**Figure A1.** Magnitude histograms as function of the ratio size $R_a$ (from left to right $R_a = 1.0, 1.7, 2.4$, respectively), and the effective area 40000 cells and 90000 cells, for upper and lower row figures respectively. The bars shows the mean histogram, and the error bars depicts the standard deviation of the twenty realizations.

34

**Fig. 2.**

---

## Author Comment (AC2) · 7 Oct 2020

Dear reviewer,

Thanks a lot for your valuable time and comments that help us to improve the manuscript. We have substantially revised the paper following your recommendations. In the next pages, we answer all your questions. We chose the following format to answer your questions:

Question/comment from the reviewer (in bold)

[Figure]

Lines in the manuscript where the answer is addressed (in red color)

Answers or replies from the authors (no special format)

New paragraphs added to the manuscript (in italic font)

We are at your disposal for any further information and well satisfied after improving our manuscript by adding new plots, figures, and references. These changes are commented below, in our reply.

Kind regards, Marisol Monterrubio-Velasco and coauthors.
* * *
1. The motivation in the Introduction was not clear. It is pretty odd to have such a long paragraph, with inserting bullet points on specific regional settings. It is very difficult to get the major points of this study, and what might be innovative compared with previous approaches that had been carried out in the same region. I feel that rewriting the Introduction is necessary, and recommend the following structure based on my reading. You may start from the significant of G-R law on probabilistic seismic hazard analysis, but it is necessary to highlight why synthetic seismicity distribution is important (the major point to deliver in this study).

→ see Introduction

We have restructured the introduction following the reviewer's comments, and now explicitly clarify the work's motivations at the end.
* * *
2. For a region that has very few earthquakes, it is straightforward to produce synthetic seismic distribution. But for the Mexican subduction zone where earthquakes are frequent, with recent great earthquakes (M>8), why is it significant and necessary to conduct such synthetic GR law analysis ?

→ lines 126 - 133:

In the lines 126 - 133 of the revised manuscript, we have answered this reviewer's question.

In this work, we focus on the Guerrero-Oaxaca (SUB3) region, because this region provides an ideal setting for testing the single asperity paradigm with the aid of TREMOL. Moreover, the quality of the database allows us to validate our code, giving support to the extension of our numerical experiments to other regions where few registered earthquakes due to scarce seismic networks. In this sense, our study pretends to be useful to generate synthetic seismicity that allows completing databases in order to carry out more accurate PSHA studies. Also we consider that our study could be appropriate to study different configuration of seismic scenarios, for example occurrence of large past events without records, or future events with a significant hazard as the case of the Guerrero gap.
* * *
3. Then you can highlight the importance of asperities and the effects of aspect ratios of asperities on GR law. If it has not been considered in previous synthetic seismicity generators, it serves as a natural innovative point of this study.

→ line 124 - 125

Thank you for the recommendation, we now have stressed the contribution of our model in this field in the referred lines.
* * *
4. It will be very helpful to explain how the asperities were identified in different subduction zones that were mentioned in lines 35-40

→ lines 104-107

Somerville et al (1999) defined an asperity as a region on the fault rupture plane that

has large slip and strength relative to the average slip on the fault. An asperity encloses fault elements in which slip is 1.5 or more times larger than the average slip. The revised version includes a similar explanation in lines 100-106 (pasted below).

"Following Somerville (1999) asperities are defined as regions of irregular shape on the rupture plane at which slip is 1.5 or more times larger than the average slip. Accordingly, Rodríguez et al., (2018) use finite-fault solutions reported for the Mexican subduction zone to estimate effective dimensions, average displacement, the combined asperity area to effective rupture area ratio, among other parameters."
* * *
5. Why do they have anything to do with TREMOL, which was first introduced in line 41? The statement here was really disjointed and should be rewritten.

We rewrite this phrase _________________________________________________________

6. Overall the grammar in the manuscript is OK, but there are lots of fragmented and/or long sentences, which are sometimes awkward to read. For example, the first sentence in section 2 contains "low magnitude previous events", which should be "previous low-magnitude events". Many other words share the similar problems and need a thorough editing work.

Thanks, we have carefully revised the grammar throughout the whole manuscript.
* * *
7. Some texts/paragraphs should belong to Introduction or Discussion, but were mixed in different sections. Here I listed a few, but not all of them. Line 59-61: statement of asperities should be moved into Introduction. Some descriptions in section 3 could be moved into Introduction as well.

Thanks again, we have moved and rearranged some paragraphs accordingly, and believe that the new version considers your comments.

────────────────────────────────────────────────────

8. For the results, I think the implications of Figure 12 are important on earthquake physics. What are the potential underlying physics that lead to such transition of GR distribution to a more characteristic earthquake pattern?

→ please see lines 362 - 375.

We added a paragraph including a possible explanation related to what we observe in the numerical results and the real seismicity behavior.

────────────────────────────────────────────────────

9. Figure 2-4 can be either merged together, or grouped together to better illustrate the locations of observed seismicity.

Done. Figs 2-4 have been grouped together in the new and self explanatory Fig. 1

────────────────────────────────────────────────────

10. I suggest deleting "for any type of" in the first sentence of the abstract.

Done.

Please also note the supplement to this comment:
https://gmd.copernicus.org/preprints/gmd-2020-115/gmd-2020-115-AC2-
supplement.pdf

──────────────────────────────────

---

## Author Response (AR2)

Dear RC1 reviewer,

First of all, we want to thank the reviewer again for his/her valuable time and comments that helped us to improve the manuscript. We tried to addressed his/her corrections in what follows. In the next pages, we answer all your questions. We have adopted the following format:

- Question/comment from the reviewer (no special format)
- Lines in the manuscript where the answer is addressed (in blue color)
- Answers or replies from the authors (in **bold**)

We are at your disposal to provide any further information you may request, and well satisfied after adding to our manuscript all the new plots, figures, and bibliography files, that are detailed in this reply.

Kind regards,

Marisol Monterrubio-Velasco and coauthors.

1. Line 124-125: There were a few papers that considered the frequency-magnitude distribution based on the dynamic rupture simulations and stochastic fault properties and stresses. For example, Ampuero, J. P., J. Ripperger and P. Mai (2006). "Properties of dynamic earthquake ruptures with heterogeneous stress drop." Earthquakes: Radiated energy and the physics of faulting: 255-261. I suggest changing this sentence.
   To our knowledge the present work is the first stochastic model based on Fiber Bundle approach that simulates the frequency-magnitude distribution and its likely dependence on the source aspect-ratio.
   **Thank you for the given references. We modified our sentence accordingly.**

2. Line 187: It is better to cite Aki (2002), rather than Leonard (2010). The moment magnitude scale was proposed a long time ago and is widely used in seismology, see $https://en.wikipedia.org/wiki/Moment_magnitude_scale$. Aki, K. and P. G. Richards (2002). Quantitative seismology, University Science Books.
   **We modified the reference. Thank you for your clarification.**

3. In Figure A2, the variable "Rec" was not defined in the main text. I suppose it is the same as Ra, right?
   **Yes you are right, we corrected the legend**

4. Line 372-375: "As Ra increases, the quantity of load that dissipates through the boundary increases." For larger Ra, more energies are dissipated by the "boundary", does that mean the ruptures of large Ra are more likely to be stopped and forms smaller events? To me, the current explanation does not make sense intuitively. If this consequence is caused by their model simplifying assumption, they shall further clarify it.
   From these results we observe that the TREMOL seismicity is highly sensible to the aspect-ratio, so tuning this parameter we can obtain either a GR-type or characteristic-type distribution. It is likely that the larger the Ra value, more energy is dissipated at the boundary, so less energy is available for secondary ruptures, either large or small ruptures are thus inhibited.
   **We observe that the model is highly sensitive to the aspect ratio. What we observe in our model is that the load dissipated through the boundary produces a large difference in the seismicity curves, from a GR-type distribution to a characteristic type distribution. It is likely that the larger the Ra value, more energy is dissipated at the boundary, so less energy is available for secondary ruptures, either large or small ruptures are thus inhibited.**

Dear RC2 reviewer,

First of all, we want to thank the reviewer again for his/her valuable time and comments that helped us to improve the manuscript. We tried to address his/her corrections. In the next pages, we answer all your questions. We have adopted the following format in our answers:

- Question/comment from the reviewer (no special format)
- Lines in the manuscript where the answer is addressed (in red color)
- Answers or replies from the authors (in **bold**)

We are at your disposal to provide any further information you may request, and well satisfied after adding to our manuscript all the new plots, figures, and bibliography files, that are detailed in this reply.

Kind regards,

Marisol Monterrubio-Velasco and coauthors.

1. 253-254: the statement is confusing and inaccurate. The fault length can be very long, e.g. San Andreas Fault, but the seismogenic depth (width) is limited by the brittle-ductile transition. An important question and reality is that no earthquakes can rupture an entire long fault system, therefore the actual ruptured aspect ratio is limited. The length/width aspect width of fault, however, can be very large and not be limited by 1 to 5.

   Based on observations Weng and Yang (2017) analyzed and reported different aspect-ratio values for strike-slip and dip-slip earthquakes. In general the aspect-ratio values are in the interval $1 < \chi < 8$, excepting for some strike-slip events that could reach larger values $\approx 40$. In our work we refer to the aspect ratio of the asperity within a fault, not the proper fault geometry. It is still true that the asperity might grow in length as opposed to width for near vertical faults once they reach the brittle limit of the crust, however we assume that this would be an extreme case.

   **Thank you for your comment. We upgraded the reference, including the results and observations from Weng and Yang, (2017). Moreover we included a brief paragraph emphasising the meaning of the aspect ratio parameter in our work.**

2. Line 559: The quantity of energy inside the seismogenic zone is lower as Ra increases. What quantity of energy? The potential energy on the fault certainly increases with the fault area, i.e. Ra in this model. This is a problematic statement. Moreover, as $Ra$ increases, the quantity of load that dissipates through the boundary increases because a larger number of cells lay in the frontier. Consequently, in the model, the seismicity distribution is clearly related to the aspect ratio of the simulated seismogenic region. As $Ra$ increases the system reduces the generation of a wide range of magnitude values, until it reaches a critical $Ra$ value $Ra \approx 2$ ($\chi \approx 4$), where the system is only able to generate very few but large earthquakes.
   **Yes, the referee is right and the statement is not properly described. In the model, the "load" mimics the energy of the system but not any physical dimension. Just to clarify that $Ra$ is not the fault area, it is the aspect ratio parameter of the asperity. In all the cases the fault area remains always constant regardless the $Ra$ value.**

3. While it is well known that asperities play significant roles in earthquakes, the asperities themselves can be heterogeneous in term of initial stress or strength distribution, as indicated by geodetic locking distribution and a number of other observations. In the simulation, I think the heterogeneity was implicitly included by different cell conditions that may evolve over time. The characteristic (or largest magnitude) earthquake in an asperity may depend on the nucleation location, as pointed by recent numerical rupture studies with constraints from geodetic locking distribution.

It would be interesting to see whether the nucleation locations play similar roles in such seismicity simulation, which can expand the discussion of this study. (Yang, H., S. Yao, B. He, and A. Newman (2019), Earthquake rupture dependence on hypocentral location along the Nicoya Peninsula subduction megathrust, Earth Plane. Sci. Lett., 520, p.10-17, $https://doi.org/10.1016/j.epsl.2019.05.030$)

**The stochastic nature of our model generates random nucleation locations for that reason we realize many simulations to statistically quantify different outputs and to analyze the model behavior. However, further experiments are required to specifically analyze the influence on the nucleation location with the magnitude.**

4. Minor points: There is a full stop in the title, very rare in scientific publications. In addition, the version of the software is very specific and not necessary. The authors may consider modifying the title into something like "Synthetic seismicity distribution in Guerrero-Oaxaca subduction zone, Mexico and its implications on the role of asperities in Gutenberg-Richter Law"

**Thank you for the recommendation, yes we modified the title**

5. The manuscript is well written in general, but there are certain typos and grammatical mistakes. I listed a few below, and expect the authors to proof read the manuscript text to fix all of them. The following line numbers refer to the track-changes version. Line 248: it is appears to Line 249: roughly square with and aspect ratio

**Thank you for the revision. We read the manuscript trying to correct the typos. We are not English native speakers, however, we done our best effort to improve the grammatical mistakes.**

[revised manuscript text omitted]

---

## Author Response (AR3)

Dear Editor Thomas Poulet and Reviewers,

We kindly regard your comments that have been improved our manuscript.
Line 384: the sentence was rephrased

Marisol Monterrubio-Velasco and coauthors.

[revised manuscript text omitted]